# Reinforcement Learning for Out-of-Distribution Reasoning in LLMs: An Empirical Study on Diagnosis-Related Group Coding

Hanyin Wang[1,2]    Zhenbang Wu[2]    Gururaj Kolar[3]    Hariprasad Korsapati[1]
Brian Bartlett[1]    Bryan Hull[4]    Jimeng Sun[2,5]
[1]Mayo Clinic Health System    [2]School of Computing and Data Science, UIUC
[3]Mayo Clinic Rochester    [4]Mayo Clinic Phoenix    [5]Carle Illinois College of Medicine, UIUC
wang.hanyin@mayo.edu, jimeng@illinois.edu

## Abstract

Diagnosis-Related Group (DRG) codes are essential for hospital reimbursement and operations but require labor-intensive assignment. Large Language Models (LLMs) struggle with DRG coding due to the out-of-distribution (OOD) nature of the task: pretraining corpora rarely contain private clinical or billing data. We introduce DRG-SAPPHIRE, which uses large-scale reinforcement learning (RL) for automated DRG coding from clinical notes. Built on Qwen2.5-7B and trained with Group Relative Policy Optimization (GRPO) using rule-based rewards, DRG-SAPPHIRE introduces a series of RL enhancements to address domain-specific challenges not seen in previous mathematical tasks. Our model **achieves state-of-the-art accuracy** on the MIMIC-IV benchmark and generates **physician-validated reasoning** for DRG assignments, significantly enhancing explainability. Our study further sheds light on broader challenges of applying RL to knowledge-intensive, OOD tasks. We observe that **RL performance scales approximately linearly with the logarithm of the number of supervised fine-tuning (SFT) examples**, suggesting that RL effectiveness is fundamentally constrained by the domain knowledge encoded in the base model. For OOD tasks like DRG coding, strong RL performance requires sufficient knowledge infusion prior to RL. Consequently, **scaling SFT may be more effective and computationally efficient than scaling RL alone** for such tasks. [1]

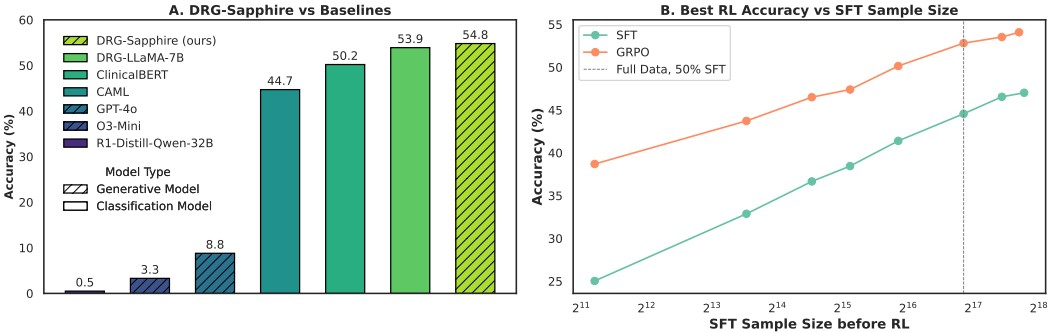

**Figure 1: Main Results.** (A) Accuracy of DRG coding on the MIMIC-IV test set (N=26,244). DRG-SAPPHIRE outperforms proprietary reasoning models and the previous SOTA model, DRG-LLaMA. Notably, classification models could not generate reasoning for DRG code assignments. (B) Best RL performance increases linearly with the logarithm of the SFT sample sizes. Dashed line marks where 50% of training data was used for SFT. Best results from vanilla GRPO runs are shown.

---

[1]Our code is available at `https://github.com/hanyin88/DRG-Sapphire`.

39th Conference on Neural Information Processing Systems (NeurIPS 2025).

# 1 Introduction

Medical codes such as DRG play pivotal roles in modern healthcare. DRG codes are fundamental to the inpatient prospective payment system, directly influencing hospital reimbursement and key quality metrics [30]. Currently, assigning DRG codes from clinical notes remains a costly and labor-intensive task, performed manually by highly trained coding specialists.

With the emergence of LLMs, there has been growing interest in leveraging these models for automated medical coding [8, 34, 38, 21, 44]. However, DRG coding remains a particularly challenging task for LLMs (Figure 1A), with prior attempts yielding limited success [38, 34]. A primary difficulty arises because DRG coding represents an **out-of-distribution (OOD)** task for off-the-shelf LLMs. Due to the private nature of medical records, most LLMs likely have minimal exposure to patient notes or billing data during pretraining. Additionally, DRG coding is inherently challenging due to: (1) a high-dimensional search space with over 700 DRG codes; (2) advanced clinical reasoning required to link diagnoses with hospital resource use and disease severity; and (3) strict hierarchical rules governing DRG assignment.

Recent advances in reasoning models, such as OpenAI-o1 [15] and DeepSeek-R1 [12], have introduced a paradigm shift in LLM post-training. By leveraging large-scale RL with verifiable rewards, these models exhibit test-time scaling through extended chain-of-thought (CoT) reasoning, achieving state-of-the-art (SOTA) performance on complex tasks like competitive mathematics. Despite this progress, the design of optimal RL algorithms for scalable training remain an open challenge [45, 24]. In the healthcare domain, RL applications using verifiable rewards are still in their early stages, with prior work primarily focused on medical knowledge benchmarks [4, 19, 20].

In this paper, we present a comprehensive exploration of large-scale, reasoning-oriented RL training for automated DRG coding from unstructured clinical notes. In theory, training towards a reasoning model is well-suited for this task: (1) it promotes the development of complex reasoning skills required for accurate code assignment; and (2) more importantly, it generates transparent rationales through CoT reasoning—a key requirement for trust and explainablity in real-world clinical applications.

Through this work, we aim to further derive insights into applying RL to challenging OOD tasks with off-the-shelf LLMs. Using Qwen2.5-7B model and GRPO with DRG-rule-based rewards, we systematically investigate the prerequisites for successful RL, the allocation of data between SFT and GRPO under a fixed data budget, and the impact of scaling SFT data. We also explore a series of RL algorithmic enhancements and adaptive learning strategies. Our core contributions are as follows:

1. We introduce DRG-SAPPHIRE, a novel model developed through large-scale RL, achieving SOTA performance in automated DRG coding. Unlike prior methods, DRG-SAPPHIRE generates clinically helpful, physician-validated reasoning, significantly improving explainability.

2. We demonstrate that the performance ceiling of RL in this OOD task is bounded by the model's capabilities before RL training. Specifically, we observe that RL performance increases linearly with the logarithm of the number of SFT examples, suggesting that scaling SFT may be more effective and computationally efficient than scaling RL alone for such tasks.

3. We propose a series of algorithmic enhancements and identify unique challenges in applying RL to DRG coding that distinguish it from mathematical domains—such as a preference for an Answer-First cognitive pattern, and sensitivity to KL divergence for stable training.

# 2 Related Work

**Automated DRG Coding**  Given their critical role in hospital operations and reimbursement, there is significant interest in automating DRG coding and enabling early DRG prediction [23, 13, 38, 10]. The prior SOTA method, DRG-LLaMA, fine-tunes a LLaMA model as a sequence classifier by replacing its generation head with a classification head [38]. Most existing approaches similarly frame DRG coding as a multi-class classification task, offering limited insight into the rationale behind code assignments. While methods like DRGCoder provide input-level weight visualizations [13], their interpretability remains insufficient for real-world clinical deployment, where transparency and explainability are critical.

**Replication Efforts of Deepseek-R1**    Recent studies have actively explored replicating the RL recipes of DeepSeek-R1, particularly in mathematical and coding domains, with varying degrees of success [47, 14, 40]. One line of work has proposed approaches to address biases and improve sample efficiency in the original GRPO algorithm [45, 24, 22]. Another active research area focuses on curriculum and staged learning strategies during reasoning-oriented RL [48, 36, 41, 16, 3].

**New Capabilities from RL?**    A central debate concerns whether RL truly fosters new capabilities beyond those already encoded in the base model. In DeepSeekMath, RL improved Majority@K but not Pass@K performance on mathematical tasks [33]. A comprehensive analysis across mathematical, coding, and visual reasoning tasks found that RL with verifiable rewards primarily reinforces existing reasoning capabilities rather than fostering novel ones [46]. Recently, Ma et al. [26] analyzed training dynamics on complex reasoning tasks, showing that RL strengthens performance within a model's existing capabilities, whereas SFT more effectively extends them beyond its current scope.

## 3    Large-scale RL for Automated DRG Coding

### 3.1    Problem Formulation

We aim to automate the hierarchical assignment of Medicare Severity Diagnosis-Related Group (MS-DRG) codes using LLMs. The MS-DRG system classifies each hospitalization into a single DRG code based on clinical complexity and resource utilization (see Appendix A.1 for details). Given a hospitalization represented by a set of clinical documents $D$, the DRG coding process applies an extraction function $h$ to identify the principal diagnosis $w_d$ or procedure $w_p$, and the presence of Complications or Comorbidities (CC) or Major Complications or Comorbidities (MCC). A hierarchical mapping function $f$ then determines the final DRG code. Formally, the MS-DRG assignment is defined as:

$$(w_d, w_p, \text{CC}, \text{MCC}) = h(D), \quad g = f(w_d, w_p, \text{CC}, \text{MCC}),$$

where $g$ is the assigned DRG code. In this paper, we use an LLM to automate this complex process.

### 3.2    Preliminary: GRPO

Compared to Proximal Policy Optimization [32], GRPO eliminates the value function and estimates the advantage using relative rewards within a group [33]. For each question $q$, GRPO samples a group of outputs $\{o_1, o_2, \cdots, o_G\}$ from the old policy $\pi_{\theta_{old}}$ and then optimizes the target policy $\pi_\theta$. In this paper, we enforce $\pi_{\theta_{old}} = \pi_\theta$ to ensure strict on-policy learning. Under this setting, we maximizing the following objective:

$$\mathcal{J}_{GRPO}(\theta) = \mathbb{E}[q \sim P(Q), \{o_i\}_{i=1}^G \sim \pi_{\theta_{old}}(O|q)]$$
$$\frac{1}{G} \sum_{i=1}^{G} \frac{1}{|o_i|} \sum_{t=1}^{|o_i|} \left[ \hat{A}_{i,t} - \beta \big( \frac{\pi_{ref}(o_{i,t}|q, o_{i,<t})}{\pi_\theta(o_{i,t}|q, o_{i,<t})} - \log \frac{\pi_{ref}(o_{i,t}|q, o_{i,<t})}{\pi_\theta(o_{i,t}|q, o_{i,<t})} - 1 \big) \right], \tag{1}$$

where $\beta$ is the coefficient for the KL divergence penalty, $\pi_{\theta_{ref}}$ is the reference policy, and $\hat{A}_{i,t}$ is the advantage, computed based on the relative rewards within each group $\{r_i\}_{i=1}^G$ as:

$$\hat{A}_{i,t} = \frac{r_i - \text{mean}(\{r_i\}_{i=1}^G)}{\text{std}(\{r_i\}_{i=1}^G)}. \tag{2}$$

Here, $r_i$ denotes the reward assigned to output $o_i$ for prompt $q$. The gradient of $\mathcal{J}_{\text{GRPO}}(\theta)$ is:

$$\nabla_\theta \mathcal{J}_{GRPO}(\theta) = \mathbb{E}[q \sim P(Q), \{o_i\}_{i=1}^G \sim \pi_{\theta_{old}}(O|q)]$$
$$\frac{1}{G} \sum_{i=1}^{G} \frac{1}{|o_i|} \sum_{t=1}^{|o_i|} \left[ \hat{A}_{i,t} + \beta \left( \frac{\pi_{ref}(o_{i,t}|o_{i,<t})}{\pi_\theta(o_{i,t}|o_{i,<t})} - 1 \right) \right] \nabla_\theta \log \pi_\theta(o_{i,t}|q, o_{i,<t}). \tag{3}$$

### 3.3    Improving GRPO Beyond the Baseline

We propose a set of strategies to address key limitations of GRPO.

**Dynamic Resampling for Advantage Preservation** Existing RL algorithms suffer from the gradient-diminishing problem. In GRPO, if all completions $\{o_i\}_{i=1}^{G}$ for a prompt $q$ receive the same reward value, the resulting advantage for this group becomes zero. As training progresses, this issue becomes more pronounced due to policy optimization and accompanying entropy collapse [45], as more prompts yield completions with no reward variance—either because all completions are perfectly correct or uniformly incorrect. This leads to a progressive decrease in the learning signal from the reward-based advantage.

To address this, we propose a **dynamic resampling** strategy (Equation 4). For each prompt $q$, if sampled completions yield zero reward variance, we resample up to $N_{\max}$ times until nonzero variance is observed. Optionally, we enforce that at least one completion receives a positive reward, guiding gradient updates toward high-reward trajectories.

$$\mathcal{J}_{GRPO}(\theta) = \mathbb{E}[q \sim P(Q), \{o_i\}_{i=1}^{G} \sim \pi_{\theta_{old}}(O|q)]$$
$$\frac{1}{G}\sum_{i=1}^{G}\frac{1}{|o_i|}\sum_{t=1}^{|o_i|}\left[\hat{A}_{i,t} - \beta\big(\frac{\pi_{ref}(o_{i,t}|q, o_{i,<t})}{\pi_\theta(o_{i,t}|q, o_{i,<t})} - \log\frac{\pi_{ref}(o_{i,t}|q, o_{i,<t})}{\pi_\theta(o_{i,t}|q, o_{i,<t})} - 1\big)\right] \quad (4)$$
$$\text{s.t.} \quad \text{Var}(\{r_i\}_{i=1}^{G}) > 0 \quad \text{within } N_{\max}, \text{ optionally: } |\{o_i \mid r_i > 0\}| > 0.$$

Our approach differs from the dynamic sampling strategy in DAPO [45], which discards prompts that yield uniformly correct or incorrect completions. Given the **data scarcity in clinical domains**, we instead maximize the utility of each training example by resampling rather than discarding.

**Intervening on Cognitive Behaviors** Cognitive behaviors, such as verification and backtracking, are critical for effective reasoning-oriented RL [11]. We explored additional reward functions and a specialized SFT dataset (detailed in Section A.3) to incentivize three cognitive patterns in CoT reasoning, as shown in Figure 2. These are: (1) **Answer-First**, where the model outputs the DRG code before CoT; (2) **CoT-First**, where the model generates CoT reasoning before the DRG code; and (3) **Differential Thinking**, where the model evaluates three potential DRG codes before selecting the most appropriate.

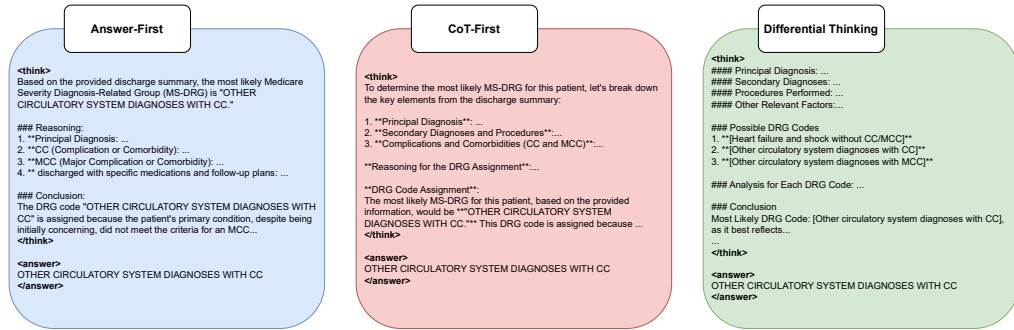

Figure 2: **Examples of Cognitive Behaviors.**

**KL Divergence Decay** The KL divergence term in the GRPO objective (Equation 1) regularizes the divergence between the target policy $\pi_\theta$ and the reference policy $\pi_{\theta_{ref}}$. However, this term exacerbates the gradient-diminishing problem in RL: as training progresses and more prompts yield zero-variance responses, the gradient, per Equation 3, becomes dominated by the KL term, pulling $\pi_\theta$ toward $\pi_{\theta_{ref}}$. This drives over-regularization toward the reference policy and risks policy degradation. Recent work suggests that removing the KL penalty enhances reasoning capabilities in mathematical domains [45, 24, 14]. Motivated by this, we explored two setups: (1) completely removing the KL divergence term from the objective, and (2) applying a cosine decay schedule to the KL term's coefficient $\beta$, smoothly reducing it to zero during training (see Section A.4 for details).

**GRPO Variants** In Equation 1, dividing by $|o_i|$ during group-level advantage normalization introduces a length bias, diminishing the influence of longer completions on the policy gradient. To address this, DAPO [45] uses $\sum_{i=1}^{G}|o_i|$ as the denominator, while Dr. GRPO [24] adopts a constant normalization factor. Additionally, Dr. GRPO removes the division by $\text{std}(\{r_i\}_{i=1}^{G})$ in Equation 2 to

mitigate question-level difficulty bias. We systematically evaluated these three strategies. Due to the strict on-policy nature of our setting ($\pi_{\theta_{\text{old}}} = \pi_\theta$), we did not explore other modifications, such as clip-higher [45].

**Reward Shaping**   We implemented two straightforward yet robust rule-based reward components: Format Reward and Accuracy Reward (detailed in the Section A.2). For the Accuracy Reward, we investigated three distinct strategies: Dense Reward, Balanced Reward, and Strict Reward. These reward functions were designed to provide varying levels of reward signal sparsity, contingent on the correctness of the DRG code, its associated principal diagnosis, and the CC/MCC status.

## 3.4   Adaptive Learning Strategy

**Curriculum Learning**   We investigate whether a curriculum learning strategy, which organizes training cases by difficulty, improves performance compared to a mixed-difficulty baseline. We evaluated four setups, detailed in Appendix A.6: (1) excluding easy cases, (2) excluding hard cases, (3) excluding both easy and hard cases (i.e., using only medium-difficulty cases), and (4) training on easy cases first, then progressing to hard cases.

**Staged Learning**   Lastly, we explored a staged learning strategy with three training phases of roughly equal length. After each phase, we identified easy and hard cases and evaluated two approaches: (1) additional SFT on hard cases, and (2) additional DPO on hard cases, before advancing to the next stage. As detailed in Appendix A.7, these approaches aim to improve the model's handling of challenging cases through targeted learning.

## 4   Implementation Details

**Dataset**   We utilized the *DRG-LLaMA* training and test sets [38], derived from the publicly available MIMIC-IV dataset of real-world medical records [17]. The full training and test sets include 236,192 and 26,244 cases, respectively. Each case uses the "brief hospital course" section from the discharge summary as input, with MS-DRG codes consolidated to version 34.0.

**Training Pipeline and Scaling Strategy**   An overview of the training pipeline is shown in Figure 3. We first sampled a reduced dataset termed **DRG-Small**, comprising 20% of the full data (N=46,758). This subset served as the foundation for extensive experiments on methodological variants and SFT–RL data mixtures, as detailed in Sections 5.2 through 5.3. After identifying the optimal configuration, we scaled training to the full dataset to produce the final DRG-SAPPHIRE model.

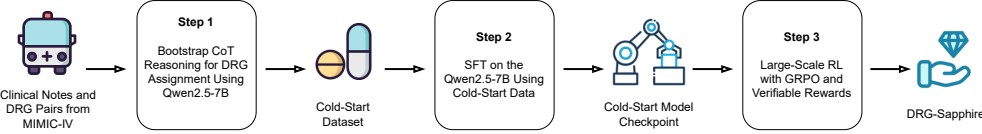

**Figure 3: Overview of Pipeline.** We construct a CoT cold-start dataset using Qwen2.5-7B-Instruct, followed by SFT with this dataset and large-scale GRPO.

**Construction of SFT Dataset**   We prompted the Qwen2.5-7B-Instruct model with medical records and ground-truth DRG codes, tasking it to generate reasoning for DRG assignments (prompt provided in Section I). After extensive prompt engineering, manual inspection by domain expert revealed that the dataset exhibits strong reasoning logic (e.g., analyzing principal diagnosis first) but frequently contains factual errors (e.g., misclassifying a condition's CC/MCC status). We also included the complete list of original V34.0 MS-DRG codes in a question–answer format within the SFT dataset.

**Model and RL Training**   We selected Qwen2.5-7B-Instructs [43] for the main experiments after evaluating various model size. GRPO training was conducted using the TRL package [37] for one epoch across all experiments.

**Evaluation Metrics**   We report model performance on the full test set using Pass@1, Pass@8, and Majority@8 (Maj@8), following prior work in reasoning-oriented RL [33, 46]. Pass@1, reported as

the model's accuracy, is the mean accuracy across eight runs. Pass@8 assesses whether the correct DRG code appears among eight generated outputs, while Maj@8 determines if the most frequent output matches the correct DRG code.

# 5 Experiments

## 5.1 Results of DRG-SAPPHIRE

Our best DRG-SAPPHIRE model was trained using a 90% SFT and 10% RL ratio on the full dataset (see Section 5.2 for SFT vs. RL ratio experiments), incorporating optimal GRPO enhancements and adaptive learning strategies (see Section 5.3 for ablation studies).

**Comparison with Baselines** As shown in Figure 1A, DRG-SAPPHIRE significantly outperforms proprietary reasoning models, non-reasoning models, and the DeepSeek-distilled Qwen 32B. It achieves new SOTA performance on DRG coding, surpassing the previous best, DRG-LLaMA-7B (54.8% vs. 53.9%). In addition to improved accuracy, DRG-SAPPHIRE provides interpretable reasoning—a compelling advantage over prior models trained purely as classifiers.

**Expert Reader Study Results** Four physicians in hospital leadership roles, actively engaged in DRG-related initiatives (e.g., reducing geometric mean length of stay), evaluated DRG-SAPPHIRE 's reasoning across 30 cases. On the dimensions of **Helpfulness** and **Accuracy**, DRG-SAPPHIRE received a median rating of 4 out of 5, suggesting good potential for real-world applications (Figure 4). Qualitative assessments highlighted the explainability of DRG coding as highly valuable for DRG-related initiatives (see Section D.1 for details), despite occasional factual inaccuracies in the reasoning.

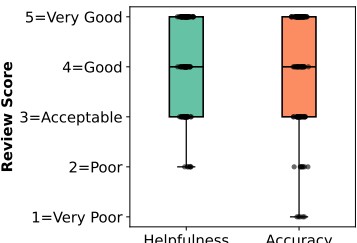

**Figure 4: Expert Reader Study.**

## 5.2 Optimizing Data Allocation Between SFT and GRPO

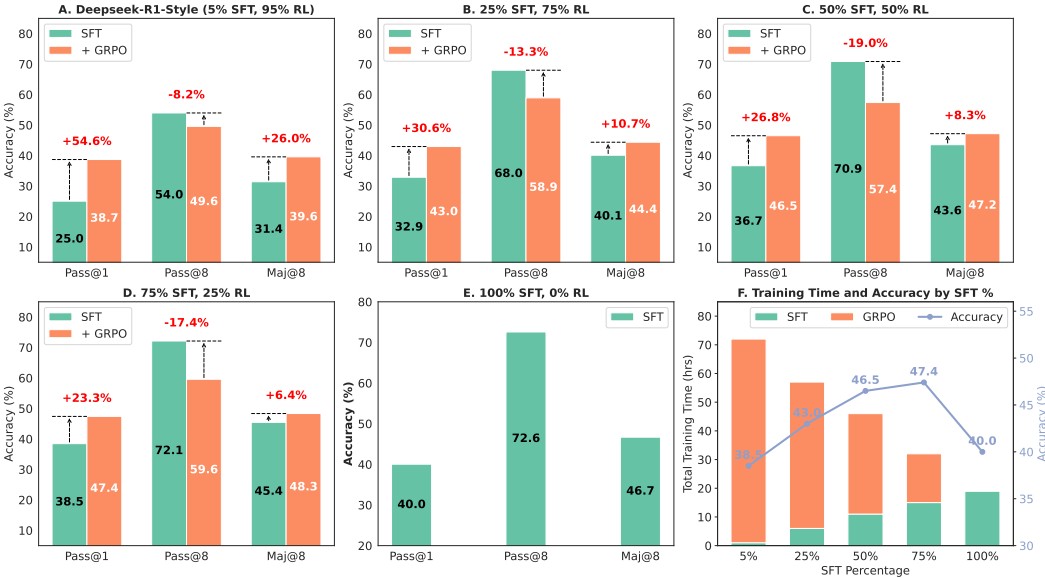

**Figure 5: Impact of SFT-GRPO Data Ratios on DRG-Small Subset.** (A–E) GRPO consistently improves Pass@1 and Maj@8 across all SFT ratios but reduces Pass@8. (F) Total training time decreases with higher SFT ratios, as GRPO is more time-consuming.

**Effect of SFT-GRPO Ratios on DRG-Small** First, we investigated the impact of varying the allocation of a fixed data budget between SFT and GRPO on the DRG-Small subset (N=46,758). This contrasts with Deepseek-R1-style training, where only minimal SFT precedes RL. Across all

data splits, GRPO consistently improved Pass@1 over the SFT baseline by an absolute margin of approximately 10 percentage points (see Figure 5). We observed that this gain is driven by improvements in Maj@8, not Pass@8; in fact, Pass@8 declines with GRPO. This pattern suggests that **RL sharpens the model's output distribution toward higher-reward pathways**, rather than introducing new reasoning capabilities in our experiments. Notably, the decline in Pass@8 during training indicates that RL may constrain diverse reasoning exploration. These findings align with recent studies [46, 33], which question whether RL improves reasoning beyond the base model's capabilities. Moreover, the ultimate performance ceiling achievable with GRPO appears to be largely determined by the capacity of the initial SFT model; **a stronger SFT foundation generally leads to better post-GRPO results**. From a computational perspective, scaling SFT prior to RL is more efficient, as GRPO entails substantial inference-time cost (see Figure 5F).

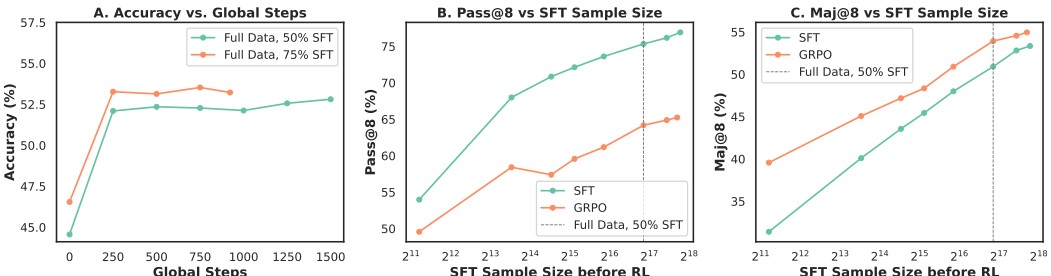

**Figure 6: Results on Full Dataset.** (A) Accuracy from the two longest GRPO runs. (B–C) Pass@8 and Maj@8 vs. SFT size. Dashed line marks where 50% of training data was used for SFT. Best results from vanilla GRPO runs are shown.

**Log-Linear Scaling of GRPO with Increasing SFT** Next, we scaled our training pipeline to the full dataset (N=236,192). Based on the results above, we started with an SFT-GRPO data ratio of 50%-50% and progressively increased the SFT ratio under a fixed total data budget. Plotting these results alongside the DRG-Small subset revealed that both GRPO and SFT performance scale approximately linearly with the logarithm of the number of SFT examples (Figure 1B). Although the number of GRPO steps varies across experiments in Figure 1B, the benefit of scaling RL appears limited in our study. Figure 6A illustrates results from our longest GRPO runs, demonstrating modest benefits beyond 500 global steps. Consistent with earlier findings, GRPO reliably improves Pass@1 and Maj@8 while reducing Pass@8 (Figure 6B and C). As the number of SFT samples increased, the slope of the GRPO curves converged toward that of SFT across all metrics. Additional results from scaling to the full dataset are detailed in Section C.3.

## 5.3 Ablation Studies on GRPO Enhancements and Adapative Learning

We present the results of ablation studies in Table 1 and Figure 12. All ablation studies were conducted on the DRG-Small dataset using Deepseek-R1-style training, with cold-start SFT on 1% of the training data (N=2,362) before RL.

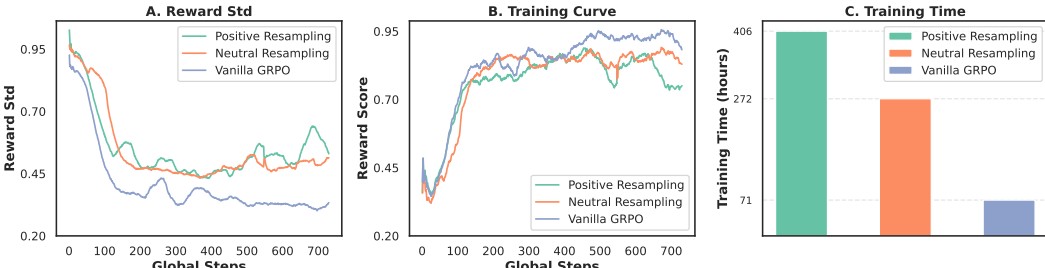

**Figure 7: Dynamic Resampling.** (A) Reward variance remains high under dynamic resampling. (B) During training, reward scores from both resampling strategies generally underperform vanilla GRPO. (C) Dynamic resampling substantially increases training time.

**Dynamic Resampling** Surprisingly, dynamic resampling—with or without a positive reward constraint—yielded marginally better or even worse performance than vanilla GRPO (Table 1), despite

| Model | DRG | | | Principal Diagnosis | | | CC/MCC | | |
|---|---|---|---|---|---|---|---|---|---|
| | Pass@1 | Pass@8 | Maj@8 | Pass@1 | Pass@8 | Maj@8 | Pass@1 | Pass@8 | Maj@8 |
| **Baseline** | | | | | | | | | |
| Vanilla GRPO + Dense Reward | 38.5 | 48.2 | 39.3 | 52.5 | 58.5 | 53.4 | 47.8 | 60.0 | 49.0 |
| **Dynamic Resampling** | | | | | | | | | |
| Neutral Resampling | 20.3 | 41.9 | 38.1 | 27.0 | 52.5 | 50.5 | 25.6 | 52.6 | 48.0 |
| Positive Reward Resampling | 39.2 | 44.8 | 39.6 | 52.9 | 56.4 | 53.3 | 48.3 | 55.6 | 49.0 |
| **Coginitive Behvaiors Intervention** | | | | | | | | | |
| COT-First | 35.5 | **52.2** | 37.4 | 50.9 | 59.6 | 52.4 | 46.3 | 66.7 | 48.4 |
| Differential Thinking | 30.2 | 47.3 | 33.9 | 46.7 | 57.0 | 50.9 | 40.6 | 63.0 | 45.2 |
| **GRPO Variants** | | | | | | | | | |
| DAPO Loss | 40.1 | 48.0 | 40.6 | **53.8** | 58.5 | **54.3** | 49.4 | 59.1 | 50.3 |
| Dr. GRPO Loss | 37.5 | 47.6 | 38.1 | 50.9 | 57.2 | 51.4 | 48.8 | 60.7 | 49.8 |
| Dr. GRPO Advantage | 38.5 | 51.9 | 39.6 | 53.4 | **60.5** | 54.3 | 47.6 | 63.6 | 49.1 |
| **KL Divergence** | | | | | | | | | |
| No KL | 39.8 | 42.4 | 39.9 | 53.6 | 55.2 | 53.7 | 49.1 | 52.3 | 49.3 |
| KL Decay | 38.2 | 42.0 | 38.3 | 52.2 | 54.7 | 52.4 | 48.8 | 53.7 | 49.0 |
| **Reward Shaping** | | | | | | | | | |
| Strict Reward | 40.1 | 49.1 | **40.9** | 52.8 | 58.1 | 53.7 | 47.6 | 59.0 | 48.8 |
| Balanced Reward | 38.1 | 51.3 | 40.0 | 52.1 | 60.4 | 53.8 | 48.2 | 64.0 | **50.7** |
| **Curriculum Learning** | | | | | | | | | |
| Remove Easy Cases | 35.8 | 51.9 | 37.6 | 50.3 | 59.2 | 51.7 | 46.6 | 65.8 | 48.7 |
| Remove Hard Cases | **40.4** | 46.6 | 40.7 | 53.2 | 56.5 | 53.7 | **49.5** | 57.2 | 50.1 |
| Remove Easy and Hard Cases | 38.7 | 48.2 | 39.4 | 52.9 | 58.1 | 53.4 | 48.3 | 59.9 | 49.3 |
| From Easy to Hard | 29.4 | 51.7 | 32.7 | 43.4 | 58.6 | 46.5 | 40.8 | **68.5** | 44.3 |
| **Staged Learning** | | | | | | | | | |
| Staged SFT | 39.3 | 49.1 | 40.0 | 52.9 | 59.2 | 53.8 | 46.0 | 58.6 | 47.1 |
| Staged DPO | 29.3 | 46.1 | 31.2 | 43.8 | 54.3 | 45.5 | 43.1 | 64.2 | 45.7 |

Table 1: **Ablation Study Results.** Rows with a blue background indicate superior Pass@1 performance compared to Vanilla GRPO + Dense Reward. Bold values denote the highest score for each metric.

preserving high reward variance (Figure 7A). More importantly, dynamic resampling proved computationally inefficient due to the frequent need to regenerate responses (Figure 7C). We hypothesize that dynamic resampling may introduces training instability by oversampling zero-variance prompts, thereby skewing batches toward OOD responses rarely generated by the current policy. Additionally, this approach may inadvertently over-penalize low-reward outputs newly introduced into the batch, further distorting the learning signal.

**Intervening on Cognitive Behaviors**   Our SFT dataset includes diverse reasoning styles, notably both Answer-First and CoT-First patterns. Interestingly, during training, the policy frequently converged toward the Answer-First strategy. To encourage CoT-First behavior, we experimented with an additional rule-based reward and adjusted the SFT dataset to explicitly promote Differential-Thinking. Although both interventions successfully elicited the intended cognitive behaviors, their performance lagged behind the naturally emerging Answer-First pattern (Table 1). This finding is surprising, as CoT-First strategies are often effective in complex reasoning tasks [39]. We hypothesize that DRG coding benefits from a direct prediction strategy, where outputting the DRG code first leverages implicit knowledge in the model's latent space, outperforming explicit CoT-grounded reasoning. These findings also align with recent studies [27, 5], which suggest that CoT and extended reasoning may not always be necessary for reasoning models, and a "no-thinking" pattern can sometimes yield better performance.

**KL Divergence**   In our experiments, **removing the KL penalty frequently led to model collapse** (Figure 8A). This contrasts sharply with findings in mathematical reasoning tasks, where the KL term is less critical, underscoring its importance for cross-domain generalization. However, in cases where training successfully completed without the KL penalty, performance surpassed that of vanilla GRPO (Table 1). Additionally, a cosine KL decay schedule appeared beneficial. While it yielded no significant gains

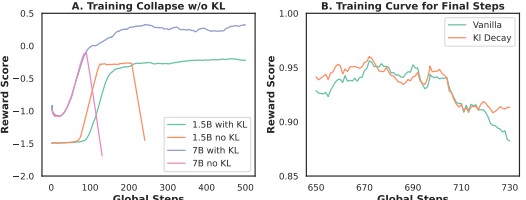

Figure 8: **KL divergence.** (A) Examples of training collapse when removing the KL divergence. (B) KL decay appears beneficial late in training.

in small-scale runs, it improved the training curve toward the end, suggesting that a lower KL penalty in later stages may help prevent over-regularization to the reference policy (Figure 8B). Indeed, KL decay proved beneficial when scaling training on the full dataset, as shown in Table 2.

**GRPO Variants**  Among three GRPO variants, the DAPO loss achieved the highest performance, while the Dr. GRPO loss performed the lowest (Table 1). This finding aligns with recent work reporting that Dr. GRPO does not outperform vanilla GRPO [6]. Across all settings, we observed **completion length contraction**: as accuracy improved, output lengths sharply decreased before stabilizing (Figure 9B). This contrasts with trends observed in mathematical tasks, where longer completions are often associated with better performance.

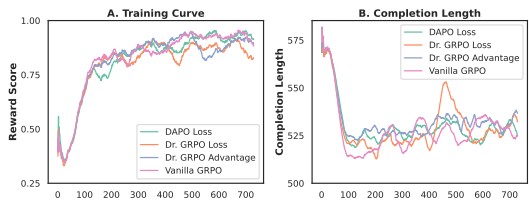

**Figure 9: GRPO Variants.** (A) Dr. GRPO loss underperforms other GRPO variants from training curve. (B) All GRPO variants exhibit similar completion length contraction.

**Reward Shaping**  The strict accuracy reward, despite providing the sparsest reward signals, outperformed both dense and balanced reward variants (Table 1). Notably, we observed no improvement in pincipal diagnosis or CC/MCC accuracy under the denser reward schemes. We hypothesize that denser rewards may lead the policy to converge prematurely to local optima, trading off global performance for easier-to-optimize intermediate signals.

**Adaptive Learning**  We observed benefits from removing easy and hard cases during training (Table 1). Similarly, recent studies in the math domain suggest that maintaining medium-level difficulty cases may be most effective for RL training [36, 41, 16, 42]. Staged learning with SFT resulted in only modest performance gains despite additional compute.

### 5.4  Prerequisites for Effective GRPO Training

We found that vanilla Qwen2.5 models (base and instruct) without SFT failed to generate correct DRG codes using GRPO alone, despite rapidly adopting the target reasoning format (Figure 10A). Post-SFT, all models showed improved RL performance that generally scaled with model size, though gains from 7B to 14B were modest (Figure 10B). Higher SFT learning rates (up to $4 \times 10^{-5}$) and extended training epochs both improved GRPO performance, though gains from additional epochs diminished at higher learning rates (Figure 10C). These results align with recent findings [29] emphasizing the importance of aggressive SFT for reasoning-intensive tasks.

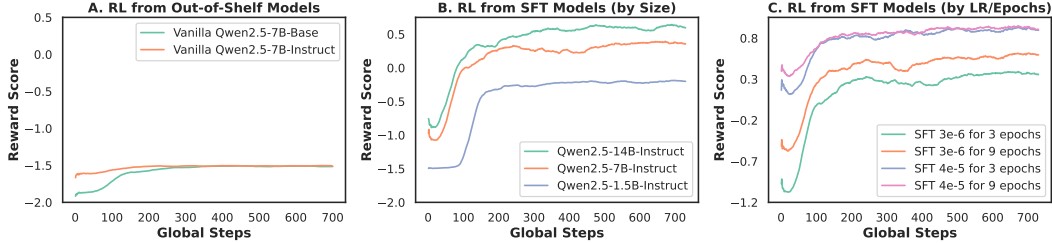

**Figure 10: Prerequisites for GRPO Training**. (A) Vanilla models without SFT fail to explore effectively, rarely generating correct DRG codes to receive positive reward signals. (B) GRPO performance increases with model size post-SFT. (C) Higher SFT learning rates and extended training epochs boost GRPO performance.

## 6  Conclusion

In this work, we used DRG coding as an empirical study to explore RL for OOD reasoning in LLMs. Our approach, applying GRPO with verifiable rewards, achieved a new SOTA performance while offering a key advantage over prior methods: the generation of physician-validated explanations through CoT reasoning. Critically, our findings reveal that RL performance on this OOD task is

fundamentally constrained by the base model's capacity prior to RL. We observed a logarithmic scaling relationship between the number of SFT examples and subsequent RL performance.

Following the successes of reasoning models like DeepSeek-R1, a prevailing narrative has been to "scale RL," leaving a critical question unanswered: what, precisely, should be scaled? Our work addresses this for complex, OOD tasks where knowledge infusion emerges as a critical component. We find that scaling SFT can be more effective and computationally efficient than scaling RL alone. Moreover, despite extensive experimentation with RL algorithmic enhancements and adaptive learning strategies, these refinements yield only modest gains compared to simply initializing RL from stronger SFT baselines—highlighting a "bitter lesson" in applying RL to tasks that fall outside the pretraining distribution of LLMs.

## 7 Acknowledgments

This research was supported by NSF awards SCH-2205289. The funder played no role in the study design, data collection, analysis, and interpretation of data, or the writing of this manuscript. This research was supported (in part) by the Mayo Clinic Clinical Practice Innovation Program. This work was also supported (in part) by the Mayo Clinic Robert D. and Patricia E. Kern Center for the Science of Health Care Delivery.

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

# Appendix

## Contents

# A Addtional Methods

## A.1 Problem Definition of MS-DRG Coding

Under the Medicare Severity DRG (MS-DRG) system, each hospitalization is assigned a single DRG code based on clinical complexity and resource utilization, following rules established by the Centers for Medicare & Medicaid Services (CMS) [7]. Given a hospital stay $D = \{d_1, d_2, \ldots, d_n\}$, where each $d_i$ represents a clinical document generated during the hospitalization, the DRG assignment process performed by human coders can be mathematically represented as follows:

1. **Extraction of Diagnoses and Procedures.** From $D$, extract a set $W = \{w_1, w_2, \ldots, w_m\}$, where each $w_i \in W$ corresponds to a distinct medical diagnosis managed or a procedure performed during the hospital stay.

2. **Identification of Principal Diagnosis or Procedure.** Select a principal diagnosis $w_d \in W$ (for medical DRGs) or a principal procedure $w_p \in W$ (for surgical DRGs), representing the main reason for admission or the primary surgical intervention. Only one—diagnosis or procedure—is designated as principal depending on the case type.

3. **Detection of Complications and Comorbidities.** Identify the presence of Complications or Comorbidities (CC) and Major Complications or Comorbidities (MCC) within $W$, forming subsets:
$$CC \subseteq W, \quad MCC \subseteq W, \quad CC \cap MCC = \emptyset,$$
which reflect distinct levels of clinical severity and resource impact.

4. **Hierarchical Mapping to DRG.** The final MS-DRG code $g$ is determined via:
$$(w_d, w_p, CC, MCC) = h(D), \quad g = f(w_d, w_p, CC, MCC),$$
where $h$ extracts the principal diagnosis or procedure and CC/MCC from $D$, and $f$ represents the CMS-defined DRG mapping logic.

## A.2 Rule-Based Reward Modeling

We adopted the following two simple yet rigorous rule-based reward components.

**Format Reward.** The Format Reward enforces a structured response, requiring reasoning content to be enclosed within `<think></think>` tags and the final answer (DRG code) within `<answer></answer>` tags. The reward is defined as:

$$S_{\text{format}} = \begin{cases} 0, & \text{if the response format is correct} \\ -2, & \text{otherwise} \end{cases}$$

**Accuracy Reward.** The Accuracy Reward evaluates the correctness of the DRG code, and applied only if the Format Reward condition is satisfied. We explored three reward shaping strategies:

### (a) Dense Reward

$$S_{\text{dense}} = \begin{cases} 2, & \text{if full match} \\ 1.5, & \text{if principal diagnosis match only} \\ 0.5, & \text{if CC/MCC match only} \\ -0.5, & \text{if valid DRG but no partial match} \\ -1.5, & \text{if invalid DRG} \end{cases}$$

### (b) Balanced Reward

$$S_{\text{balanced}} = \begin{cases} 2, & \text{if full match} \\ 1, & \text{if principal diagnosis match only} \\ 1, & \text{if CC/MCC match only} \\ -0.5, & \text{if valid DRG but no partial match} \\ -1.5, & \text{if invalid DRG} \end{cases}$$

**(c) Strict Reward**

$$S_{\text{strict}} = \begin{cases} 2, & \text{if full match} \\ 0, & \text{if partial or no match but valid DRG} \\ -1.5, & \text{if invalid DRG} \end{cases}$$

## A.3 Enforcing Cognitive Behaviors

To incentivize CoT-first cognitive behaviors, we introduced an additional format penalty. If the model outputs a DRG code within the first 50 tokens of the reasoning, a penalty score of $-0.5$ is assigned.

To encourage differential thinking, we reconstructed the SFT dataset using the same data. We designed a new prompt for the Qwen2.5-7B-Instruct model to generate three potential DRG codes per case (prompt provided in Section I), each accompanied by reasoning, before selecting the most appropriate DRG code.

## A.4 KL Divergence Decay

To gradually relax the regularization imposed by the KL penalty, we apply a cosine decay to the KL coefficient $\beta$, reducing it from its initial value to zero over the course of training. For global step $t$ and total training steps $T$, the decay factor is defined as:

$$\text{decay\_factor}(t) = 0.5 \cdot \left(1 + \cos\left(\frac{\pi t}{T}\right)\right)$$

The decayed coefficient at step $t$ is then:

$$\beta_t = \beta \cdot \text{decay\_factor}(t)$$

This decay schedule promotes stability in the early stages of training while encouraging exploration in later updates.

## A.5 GRPO Variants

We implemented the loss functions for different GRPO variants as follows:

```
#Vanilla GRPO loss. This is the original implementation in TRL v0.15.1.
vanila_GRPO_loss = ((per_token_loss * completion_mask).sum(dim=1) / completion_mask.
    sum(dim=1)).mean()

#DAPO loss. This is the default implementation in TRL since v0.16.0.
DAPO_GRPO_loss = (per_token_loss * completion_mask).sum() / completion_mask.sum()

#Dr.GRPO loss. We set max_tokens to 1024.
max_tokens = 1024
Dr_GRPO_loss = ((per_token_loss * completion_mask).sum(dim=1) / max_tokens).mean()
```

For the Dr. GRPO advantage, we modified the advantage computation by removing the denominator in Equation 2.

## A.6 Curriculum Learning

Unlike mathematical problems, the difficulty of DRG coding is not easily defined. High prediction accuracy does not necessarily indicate that a DRG code is inherently easy. For instance, the frequently occurring code "Septicemia or Severe Sepsis without MV >96 Hours with MCC" may be straightforward in most cases but can become challenging when the clinical narrative emphasizes a different primary condition, such as a urinary tract infection. Moreover, no standardized benchmark exists to quantify DRG coding difficulty.

To address this, we employed a static online filtering strategy. For each experiment, we first ran the model without filtering to establish a baseline. Easy cases were defined as those with zero reward variance and perfect accuracy scores. Hard cases were defined as those with zero reward variance, an accuracy score of $-0.5$ under the dense or balanced reward, and $0$ under the strict reward. We then reran the experiment from the SFT model after excluding these filtered cases.

## A.7 Staged Learning

For staged learning, we divided the training process into three stages, each with approximately the same number of global steps. After each stage, we identified hard and easy cases using the methodology described in Section A.6. For hard cases, we prompted the most recent GRPO model checkpoint to generate reasoning given the case and the correct DRG code. We then performed SFT or DPO on this new dataset. For SFT, we used a learning rate of $4 \times 10^{-5}$ for 3 epochs. For DPO, we designated the original model output as the rejected response, and trained with a learning rate of $3 \times 10^{-6}$ for 3 epochs.

# B    Additional Implementation Details

## B.1    SFT Training Details

We used the SFT trainer from the TRL library for all SFT runs [37], with DeepSpeed ZeRO Stage 3 [31] and the AdamW optimizer [25]. Training was conducted on 4 H100 or A100 GPUs, depending on availability, using bf16 precision. We set `packing=False` and `max_seq_length` to 12846. A cosine learning rate schedule with a minimum of 10% of the initial rate was applied, along with a warm-up ratio of 0.05. The global batch size was adjusted based on VRAM constraints to roughly match the number of unique cases per step used in GRPO training.

## B.2    GRPO Training Details

Our implementation of GRPO was based on the Open R1 framework [9], which leverages vLLM [18] for inference and GRPO Trainer from TRL library (v0.15.1) [37] for training. All training was conducted on 3 to 5 H100 or A100 GPUs, depending on availability, using bf16 precision. For all GRPO experiments, we set `num_generations` to 8, `per_device_train_batch_size` to 2 or 4, and `gradient_accumulation_steps` to 32 or 64, ensuring a consistent global batch size of 512 across experiments. Each global step consisted of 64 unique prompts, each with 8 generated completions. We set the `max_prompt_length` to 4096 and the `max_completion_length` to 10240. The temperature of the policy model is set to 1. All other training parameters were kept at their default values, including a KL regularization coefficient of $\beta$= 0.04.

All GRPO experiments were run for a single epoch. As we enforced $\pi_{\theta_{\text{old}}} = \pi_\theta$ to ensure strict on-policy learning, this is equivalent to setting `num_iterations` to 1 in later versions of the TRL library. We adopted the default system prompt from Open R1.

## B.3    Experimental Hyperparameters

**SFT**    For SFT, we experimented with different learning rates and training epochs, as detailed in Section 5.4. For all experiments in Section 5.3, we initialized GRPO training with an SFT model trained using a learning rate of $4 \times 10^{-5}$ for 9 epochs. The only exception is the result shown in Figure 8A, which illustrates training collapse from earlier runs using a learning rate of $3 \times 10^{-6}$. For experiments in Sections 5.2, we used SFT models trained with a learning rate of $4 \times 10^{-5}$ but for 3 epochs, as the SFT data was scaled.

**GRPO**    For GRPO, we experimented with different learning rates and scheduling strategies, as detailed in Section C.1. For all experiments in Sections 5.2 to 5.3, we used a GRPO learning rate of $3 \times 10^{-6}$ with a constant learning rate scheduler and a warmup ratio of 0.1.

## B.4    Evaluation Details

We used vLLM [18] for inference during evaluation. All evaluations were conducted on the full test set (N = 26,244). We set the temperature to 0.6, `top_p` to 0.95, and `max_tokens` to 4096. To compute Pass@8, we set n to 8 in `SamplingParams`, generating eight completions per case. Pass@1 is reported as the mean accuracy across these eight generations. For evaluation, we extracted the DRG code enclosed within `<answer></answer>` tags and computed exact match against the reference code after text normalization. All training curves figures in Section 5.3 are smoothed using a moving average with a window of 50 steps.

## B.5 Dynamic Resampling Details

For both neutral and positive dynamic resampling, we set the maximum number of regeneration attempts to 12. During regeneration, the model randomly selects a temperature from the set $\{0.7, 0.8, 0.9, 1.0\}$.

## C Additional Results

### C.1 Experiments with GRPO Hyperparameters

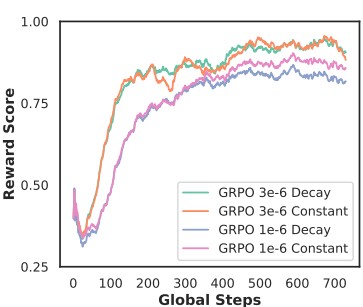

**Figure 11: Effect of Learning Rate and Scheduler on GRPO.**

We conducted a limited hyperparameter search to tune the learning rate and scheduler for GRPO. As shown in Figure 11, a learning rate of $3 \times 10^{-6}$ consistently outperformed $1 \times 10^{-6}$, yielding faster convergence and higher final reward scores. We further compared constant and decaying schedules and observed similar overall performance. Notably, the constant schedule was more effective at lower learning rates, though this advantage diminished as the rate increased. It is possible that a constant learning rate helps mitigate gradient vanishing during RL training, as discussed in Section 3.3.

### C.2 Accuracy with RL Training in Ablation Studies

We present accuracy results from various ablation studies in Section 5.3, as shown in Figure 12.

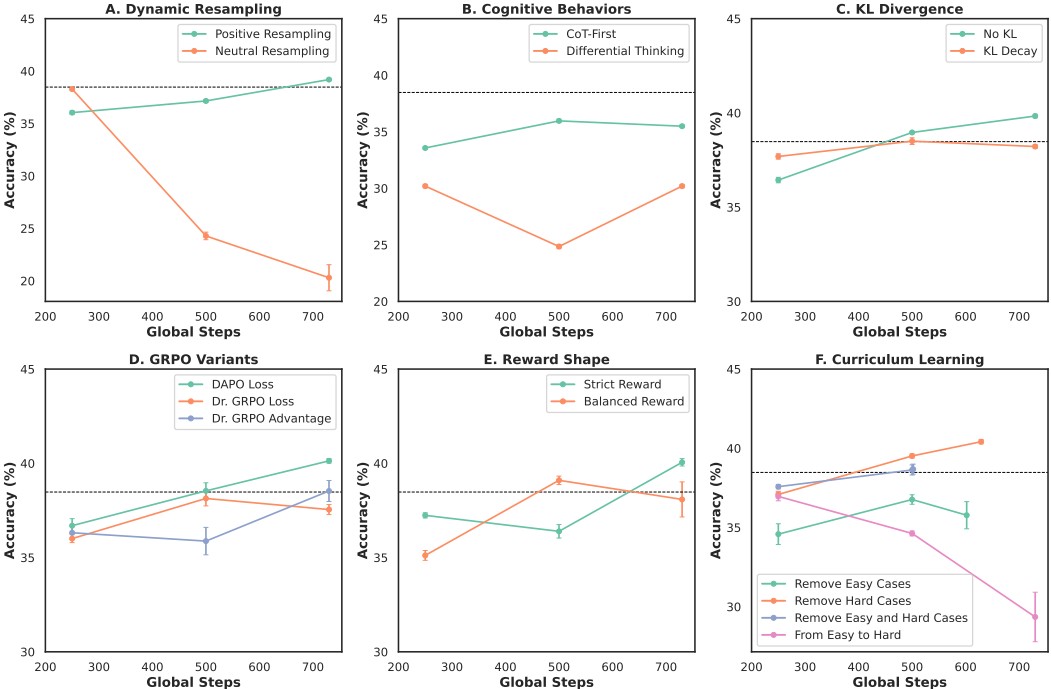

**Figure 12: Accuracy with RL Training in Ablation Studies.** The dashed line indicates the baseline performance of vanilla GRPO with dense rewards. Error bars indicate the standard deviation across 8 runs.

### C.3 Additional Results from Scaling to the Full Dataset

We present experimental results on the full dataset with varying SFT-to-RL data splits (Table 2). Our best configuration, combining DAPO loss, strict accuracy reward, and KL decay (Section 5.3), consistently outperformed vanilla GRPO across all experiments. Curriculum learning, implemented

| Model | DRG | | | Principal Diagnosis | | | CC/MCC | | |
|---|---|---|---|---|---|---|---|---|---|
| | Pass@1 | Pass@8 | Maj@8 | Pass@1 | Pass@8 | Maj@8 | Pass@1 | Pass@8 | Maj@8 |
| **50% SFT** | | | | | | | | | |
| SFT | 44.6 | 75.3 | 50.9 | 58.1 | 77.1 | 63.4 | 51.8 | 80.7 | 58.4 |
| Vanilla GRPO | 52.8 | 64.2 | 53.9 | 63.9 | 70.9 | 64.9 | 59.0 | 69.6 | 60.2 |
| Best Config | 53.7 | 59.1 | 53.9 | 63.5 | 66.9 | 63.9 | 58.8 | 64.1 | 59.4 |
| **75% SFT** | | | | | | | | | |
| SFT | 46.5 | 76.2 | 52.8 | 59.3 | 77.6 | 64.5 | 53.3 | 80.7 | 59.6 |
| Vanilla GRPO | 53.5 | 64.9 | 54.6 | 64.0 | 71.4 | 65.1 | 59.2 | 69.7 | 60.5 |
| Best Config | 54.6 | 60.6 | 54.9 | 64.4 | 68.2 | 64.8 | 59.6 | 65.3 | 60.2 |
| Best Config - KL Decay | 54.0 | 65.3 | 55.0 | 63.8 | 71.0 | 64.9 | 58.7 | 69.4 | 60.0 |
| Best Config + Remove Hard Case | 54.4 | 58.1 | 54.5 | 63.8 | 66.1 | 64.0 | 59.1 | 62.7 | 59.4 |
| Best Config + Remove Easy and Hard Case | 54.7 | 58.8 | 54.8 | **64.5** | 67.1 | 64.8 | 59.5 | 63.5 | 59.9 |
| **90% SFT** | | | | | | | | | |
| SFT* | 10.5 | 41.4 | 35.7 | 12.4 | 47.1 | 43.6 | 11.7 | 47.0 | 41.6 |
| Vanilla GRPO | 54.1 | 65.2 | 55.0 | 64.3 | 71.2 | 65.2 | 59.8 | 70.0 | 60.7 |
| Best Config | 54.6 | 62.2 | 54.9 | 64.0 | 68.9 | 64.6 | 59.3 | 66.3 | 60.0 |
| Best Config - KL Decay | 54.2 | 66.9 | **55.4** | 63.9 | 72.1 | 65.2 | 59.3 | 70.8 | 60.8 |
| Best Config + Remove Hard Case | **54.8** | 60.3 | 54.9 | 64.4 | 68.1 | 64.7 | **59.9** | 64.9 | 60.4 |
| Best Config + Remove Easy and Hard Case | 54.5 | 61.4 | 54.8 | 64.2 | 68.8 | 64.9 | 59.0 | 65.4 | 59.8 |
| **95% SFT** | | | | | | | | | |
| SFT | 47.0 | **76.9** | 53.3 | 59.7 | **78.5** | 65.0 | 53.9 | **81.0** | 59.9 |
| Vanilla GRPO | 53.5 | 67.7 | 55.2 | 64.0 | 72.6 | **65.6** | 59.2 | 72.3 | **61.0** |
| Best Config | 54.4 | 64.9 | 55.1 | 64.2 | 70.5 | 65.0 | 59.4 | 69.4 | 60.5 |
| Best Config - KL Decay | 53.0 | 69.4 | 55.1 | 63.2 | 73.5 | 65.1 | 58.4 | 73.4 | 60.8 |
| Best Config + Remove Hard Case | 54.3 | 62.8 | 54.7 | 64.2 | 69.7 | 64.9 | 59.5 | 67.4 | 60.3 |
| Best Config + Remove Easy and Hard Case | 52.9 | 69.3 | 54.9 | 63.1 | 73.0 | 65.0 | 58.1 | 73.3 | 60.5 |

**Table 2: Scaling of GRPO on the Full Dataset.** All experiments were conducted on the full training set (N=236,192) with varying SFT-to-RL ratios, and the best result from each experiment is reported in the table. The best configuration of GRPO consists of DAPO loss, strict accuracy reward, and KL decay. The row highlighted in blue indicates the best Pass@1 performance. Bold values denote the highest score for each metric. * The SFT checkpoint from the 90% SFT runs exhibited format-following instability, resulting in lower-than-expected scores. Despite this unstable SFT baseline, RL training remained robust.

by excluding hard or easy cases, further improved performance. The best overall performance of DRG-SAPPHIRE was achieved with a 90% SFT and 10% RL split using the best GRPO configuration and hard-case exclusion.

Notably, we excluded the SFT result from the 90% SFT runs and the vanilla GRPO result from the 95% SFT runs as outliers in Figure 1B and Figures 6B and C. The SFT checkpoint from the 90% SFT runs exhibited format-following instability, leading to lower-than-expected scores. Despite this unstable SFT baseline, RL training remained robust, effectively leveraging the knowledge infused through SFT and ultimately producing our best overall results. The RL results from the 95% SFT runs are likely not representative of true RL potential due to insufficient RL training ($<$ 250 global steps). Additionally, we did not conduct experiments without KL decay or with curriculum learning for the 50% SFT runs, given the limited performance observed with vanilla GRPO in that setting. Lastly, we applied early stopping for the GRPO experiments from the 50% SFT runs due to lack of improvement (Figure 6A).

### C.4 Error Analysis

We identified 9,537 cases consistently misclassified by all three checkpoints from different training stages: the cold-start SFT (SFT model in Figure 5A), the small-scale GRPO (GRPO model in Figure 5B), and the final DRG-SAPPHIRE model. As shown in Table 3, even in these challenging cases, the model improved progressively during training, becoming notably "less wrong."

| Model | % Wrong Principal Diagnosis | % Wrong CC/MCC | % Both Wrong |
|---|---|---|---|
| Cold-start SFT | 85.1% | 64.2% | 47.4% |
| Cold-start SFT + GRPO | 80.4% | 63.1% | 41.1% |
| DRG-SAPPHIRE | 74.9% | 62.7% | 33.6% |

**Table 3: Progressive error reduction across training stages on consistently misclassified cases.**

Additionally, a physician conducted a manual error analysis on 100 randomly selected cases misclassified by the final DRG-SAPPHIRE model, using the same error taxonomy as DRG-LLaMA. The results are presented in Table 4.

| Error Category | Percentage (%) |
|---|---|
| Difficulty selecting correct base DRG | 32 |
| Erroneous CC/MCC | 25 |
| Potential incorrect DRG label | 13 |
| Inadequate clinical concept extraction | 12 |
| Information needed for DRG prediction unavailable | 10 |
| Other (e.g., procedure instead of medical DRG) | 8 |

**Table 4: Physician-annotated error taxonomy for 100 misclassified cases by DRG-SAPPHIRE.**

### C.5 Additional Results on a Real-World Dataset

We evaluated DRG-SAPPHIRE on an internal dataset from a healthcare institution. This dataset was constructed via stratified random sampling of 2,500 real-world cases, with DRG codes sampled in proportion to their frequencies in the MIMIC test set to ensure comparability. DRG-SAPPHIRE achieved an accuracy of 53.6% on this internal dataset, compared to 54.8% on MIMIC.

## D  Additional Discussion

### D.1 Clinical Applications of Automated DRG Coding with Reasoning

DRG-SAPPHIRE shows good potential for real-world clinical applications. Two illustrative use cases are presented below:

1. Currently, DRGs are assigned by professional coders and are typically available only after hospital discharge. DRG-SAPPHIRE can provide early DRG predictions to inform hospital operations and financial forecasting.

2. DRG-SAPPHIRE can support DRG-related operational and quality improvement initiatives, such as those aimed at reducing the geometric length of stay, a metric directly determined by DRG. It provides transparent, interpretable explanations of DRG assignments, enabling clinicians to improve their clinical documentation to better reflect patient severity.

In discussions with domain experts, we also found that the impact of erroneous DRG code assignments is highly dependent on the intended use case. For high-stakes applications such as automated billing, the acceptable error rate must be extremely low due to potential financial implications. In contrast, for operational or educational tools—such as the one mentioned above that helps physicians understand DRG assignments—a higher degree of fault tolerance may be acceptable.

### D.2 Practical Implication of Improved Pass@1 but Not Pass@k

Our experiments demonstrate that RL improves Pass@1 (i.e., accuracy) but not Pass@k for higher k values, indicating that RL enhances the model's ability to produce the correct DRG code in a single attempt without necessarily improving its broader reasoning capacity. However, this outcome aligns well with the requirements of high-stakes clinical applications like DRG coding, where only the first prediction truly matters, as users typically do not sample multiple outputs. Moreover, selecting the correct answer from multiple candidate responses is challenging, as methods beyond the best-of-N approach, which RL already optimizes by improving Pass@1 through better majority voting (Maj@k), are not well-established.

### D.3 DRG vs ICD Coding

Although both DRG and International Classification of Diseases (ICD) codes serve clinical and administrative purposes, they differ significantly in classification approach and real-world applications.

DRG assignment is typically formulated as a multi-class classification task, in which exactly one DRG code is assigned to summarize resource utilization and clinical complexity for an entire hospitalization. In contrast, ICD coding is a multi-label classification problem, as multiple ICD codes—covering both diagnoses and procedures—may be assigned to document a single encounter. Furthermore, the two coding systems exhibit distinct hierarchical structures: DRG assignment explicitly emphasizes identifying a principal diagnosis or procedure that primarily drives the hospitalization, along with secondary conditions that influence clinical complexity and reimbursement [7]. Finally, the utilization contexts of these codes differ substantially: DRGs are directly linked to inpatient reimbursement and hospital resource management, whereas ICD codes serve broader purposes spanning both inpatient and outpatient documentation.

## E Additional Related Work

Several studies have explored constructing reward signals in RL beyond rule-based approaches. Mu et al. [28] proposed an effective method that combines fine-grained, composable rules with an LLM-based grader to form a hybrid reward signal, balancing model safety and usefulness. Sun et al. [35] introduced a framework that grounds vision-language models (VLMs) in medical knowledge through symbolic representations of clinical reasoning and employs a symbolic reward function to assess VLM outputs for correctness and clinical validity.

## F Limitations and Future Work

Our study encountered several limitations. First, we employed only rule-based rewards for final DRG assignments, without utilizing process supervision during the reasoning steps. While it is unclear how best to implement such supervision, theoretically, more granular and dense reward signals throughout the reasoning process could help guide the policy toward more effective exploration. Future work exploring this direction—potentially combining explicit DRG rules with techniques such as process reward modeling—represents an intriguing avenue.

Second, we applied relatively static curriculum learning and case-filtering strategies, which were conducted only once following the completion of a base run. A dynamic, online, difficulty-based filtering approach—applied at the per-batch level—may be more effective and warrants further investigation.

Lastly, our work focused exclusively on the challenging task of DRG coding. Extending our approach to additional medical-domain tasks and to OOD tasks across domains would be a valuable direction for future work. In particular, it would be compelling to investigate whether scaling RL across multiple tasks and domains fosters exploration of diverse reasoning trajectories beyond the base model, rather than merely refining its output distribution toward higher-reward outcomes.

## G Data Access

Access to MIMIC-IV can be requested via [1], which requires signing a data use agreement. The training and test datasets used in this study can be obtained by following the code repository provided in [38]. For experiments involving MIMIC-IV data and proprietary models, we adhered to the guidelines in [2] and utilized the Azure OpenAI service.

# H   Instruction to Reviewers

Physician reviewer instructions for scoring DRG-SAPPHIRE's reasoning traces are provided below.

---

**Instruction to Reviewers**

1. You will be provided with a discharge summary from the public MIMIC-IV dataset, along with a corresponding DRG code assignment and its rationale generated by a large language model (LLM).

2. Please note that, similar to existing DRG prediction tools currently in use, the LLM-generated DRG code assignment may be either correct or incorrect.

3. Your task is to rate the LLM output along two dimensions: **Helpfulness** and **Accuracy**, using a scale from **1 to 5 (very poor, poor, acceptable, good, or very good)**, where higher scores indicate better quality.

4. **Helpfulness**: For this dimension, please answer the question: "Is the LLM's reasoning and explanation helpful to frontline healthcare providers?" Reflect on real-world initiatives you are engaged in that center around DRG optimization (e.g., efforts to reduce geometric mean length of stay). Assess whether the information provided by the LLM would meaningfully assist physicians in such settings, addressing questions commonly raised in practice.

**Rubric:**

- Score of 1 (very poor): The content is not helpful — for example, it may be too generic, lack necessary detail, or be overly vague.
- Score of 3 (acceptable): The content is sufficiently helpful and acceptable for use in real-world clinical settings.
- Score of 5 (very helpful): The content is highly helpful and could positively impact real-world DRG-related initiatives.

5. **Accuracy**: For this dimension, please answer the question: "Does the information provided by the LLM accurately reflect MS-DRG assignment rules?" Base your evaluation on your best knowledge and understanding of the MS-DRG system.

**Rubric:**

- Score of 1 (very poor): The information is substantially inaccurate.
- Score of 3 (acceptable): The information is accurate enough to support decision-making by frontline healthcare providers.
- Score of 5 (very accurate): The information is highly accurate and consistent with MS-DRG assignment rules.

---

# I Prompts to LLM

The prompt used with Qwen2.5-7B-Instruct for generating the SFT cold-start dataset is provided below.

---

### Generate Reasoning for DRG Assignment

MS-DRG (Medicare Severity Diagnosis-Related Groups) is a system used by the U.S. Centers for Medicare & Medicaid Services to classify hospital inpatient stays for payment purposes. It groups patients with similar clinical conditions and resource needs into categories to determine reimbursement amounts. Each MS-DRG is assigned based on the patient's principal diagnosis, secondary diagnoses, procedures performed, age, discharge status, and other factors. The goal is to ensure fair and consistent hospital reimbursement based on the severity of the illness and the complexity of care required.

CC and MCC in MS-DRG:

- CC (Complication or Comorbidity): A secondary diagnosis that increases the complexity of care and resource utilization.
- MCC (Major Complication or Comorbidity): A more severe secondary condition that has a significant impact on resource use and hospital reimbursement.
- MCCs have a greater effect on the DRG weight than CCs due to increased patient care complexity.

What is the most likely Medicare Severity Diagnosis Related Group (MS-DRG) based on the following discharge summary of a hospitalization? Provide the complete text description of the DRG code without including the numerical code.

**[Discharge Summary]**

The answer is **[DRG Code]**

Explain the reason why the DRG code is assigned. In your reasoning step, assume you don't know the right DRG code yet.

---

For SFT dataset generation with differential thinking, we added the following content to the prompt above.

---

### Additional Prompt to Generate Differential Thinking

First, analyze the case details, including the principal diagnosis, secondary diagnoses, procedures performed, and any other relevant factors that may influence DRG assignment.

Next, list three possible DRG codes that you believe are the most likely. Enclose each DRG code in square brackets, like this: [Heart failure without MCC]. One of the three DRG codes should be **[DRG Code]**.

Then, provide a detailed analysis for each of the three DRG codes. For each one, explain:

- Why this DRG code might be appropriate?
- Why it might not be appropriate?

Finally, based on your analysis, identify the most likely DRG code and it should be **[DRG Code]**.

---

## J  Example Outputs from DRG-Sapphire

---

**Example One**

**<think>**
Based on the provided discharge summary, the most likely Medicare Severity Diagnosis-Related Group (MS-DRG) is **"ESOPHAGITIS GASTROENTERITIS AND MISCELLANEOUS DIGESTIVE DISORDERS WITHOUT MCC."**
Let's break down the reasoning step-by-step:

1. **Principal Diagnosis and Conditions:**
   - The principal diagnosis is colitis, which is a digestive disorder.
   - There are no complications or comorbidities (CCs or MCCs) that significantly impact the complexity of care or resource utilization more than the primary diagnosis of colitis.

2. **Secondary Conditions and CCs/MCCs:**
   - **Atypical Carcinoid Tumor of the Lung:** A significant condition, but not the primary reason for the hospitalization and does not elevate complexity of care related to colitis.
   - **Metastatic Pancreatic Neuroendocrine Tumor:** Also significant, but not directly impacting the care for colitis.
   - **Perforated Diverticulitis:** Required prior hospitalization but is not contributing to the current hospital stay.
   - **Transaminitis:** Mild and resolved quickly; not related to the colitis.
   - **Hypothyroidism:** Chronic and well-managed; no impact on the complexity of care.
   - **Ischemia:** A possible cause of colitis, but not confirmed and does not significantly impact complexity.

3. **Complications or Comorbidities (CCs):**
   - The atypical carcinoid tumor, metastatic pancreatic neuroendocrine tumor, and ischemia are significant but not classified as CCs that elevate complexity for colitis.
   - The perforated diverticulitis is historical and not relevant to the current hospital stay.

4. **Major Complication or Comorbidity (MCC):**
   - No significant conditions qualify as an MCC that would elevate the DRG weight beyond the primary diagnosis of colitis.

Given these conditions, the DRG code is assigned based on the primary diagnosis of digestive disorders, specifically colitis, without any significant complications or comorbidities that would elevate it to a higher complexity category. Therefore, the most appropriate DRG is **"ESOPHAGITIS GASTROENTERITIS AND MISCELLANEOUS DIGESTIVE DISORDERS WITHOUT MCC."**
**</think>**

**<answer>**
**ESOPHAGITIS GASTROENTERITIS AND MISCELLANEOUS DIGESTIVE DISORDERS WITHOUT MCC**
**</answer>**

---

**Example Two**

<think>
Based on the discharge summary provided, the most likely Medicare Severity Diagnosis-Related Group (MS-DRG) is **PANCREAS LIVER AND SHUNT PROCEDURES WITH MCC**.
To determine this, let's break down the key elements of the case:

1. **Principal Diagnosis and Procedures:**
   - The patient underwent a left hepatic lobectomy and placement of biliary stents.
   - The lobectomy is a major liver surgery—this alone is a significant factor for DRG assignment.

2. **Complications and Comorbidities:**
   - Post-operative bilious drainage from the JP (jejunostomy) tube.
   - Increased wound drainage requiring VAC (wound vacuum) placement.
   - ERCP performed to manage bile duct stones and issues.
   - Biliary stents placed during ERCP.
   - Elevated WBC count and suspected infection requiring IV antibiotics and infectious disease consultation.

3. **MCC Analysis:**
   - The ERCP, biliary stents, and post-op complications significantly impacted recovery.
   - These events qualify as a Major Complication or Comorbidity (MCC) under DRG criteria.

4. **Severity and Complexity:**
   - Multiple interventions (ERCP, stent placement, IV antibiotics).
   - Extended inpatient stay due to complications.
   - High resource utilization consistent with MCC-level DRG classification.

5. **Consistency with MS-DRG Criteria:**
   - MS-DRG assignment accounts for diagnosis, procedures, and complications.
   - This case aligns with the category for major liver/pancreas procedures involving MCCs.

Therefore, the DRG code **PANCREAS LIVER AND SHUNT PROCEDURES WITH MCC** is assigned because the case involves a significant liver procedure (left hepatic lobectomy) and multiple major complications and interventions. The care complexity and extended course fulfill criteria for an MCC classification.

</think>

<answer>
**PANCREAS LIVER AND SHUNT PROCEDURES WITH MCC**
</answer>

# K    Example Outputs Demonstrating Different Cognitive Behaviors

---

**Example from Answer-First**

**<think>**
Based on the provided discharge summary, the most likely Medicare Severity Diagnosis-Related Group (MS-DRG) is **"OTHER CIRCULATORY SYSTEM DIAGNOSES WITH CC."**
**Reasoning:**

1. **Principal Diagnosis:**
   - The patient presents with chest pain and ST elevations, initially suggestive of STEMI (ST-Elevation Myocardial Infarction).
   - Cardiac catheterization revealed normal coronary arteries, ruling out coronary artery disease.
   - ST elevations without coronary lesions point to myopericarditis or a related circulatory system condition.

2. **CC (Complication or Comorbidity):**
   - Chest pain, ST elevations, and treatment with ibuprofen and colchicine increased care complexity.
   - Mild echocardiographic findings (dilated aortic root and ascending aorta) required follow-up and additional monitoring, qualifying as a CC.

3. **MCC (Major Complication or Comorbidity):**
   - Despite significant diagnostic workup, no MCC-level condition was present.
   - No major complication or comorbidity substantially elevated resource use or reimbursement criteria.

4. **Discharged with Specific Medications and Follow-Up Plans:**
   - The patient was discharged on ibuprofen and colchicine.
   - A follow-up echocardiogram was planned, indicating ongoing resource use.

**Conclusion:**
The DRG code **"OTHER CIRCULATORY SYSTEM DIAGNOSES WITH CC"** is assigned because the patient's primary condition, although initially concerning, did not meet the criteria for an MCC. The presence of chest pain, ST elevations, treatment complexity, and follow-up needs justify a CC classification due to increased resource utilization and care complexity.
**</think>**

**<answer>**
**OTHER CIRCULATORY SYSTEM DIAGNOSES WITH CC**
**</answer>**

## Example from CoT-First

<think>
To determine the most likely MS-DRG for this patient, let's break down the key elements from the discharge summary:

1. **Principal Diagnosis:**
   - The patient presents with ST elevations in the EKG, initially consistent with a STEMI (ST-Elevation Myocardial Infarction).
   - Cardiac catheterization revealed normal coronaries (RRA – Right Rectus Artery), indicating a false positive STEMI.
   - The patient was diagnosed with myopericarditis, which can cause chest pain and ST segment elevations.

2. **Secondary Diagnoses and Procedures:**
   - Myopericarditis (secondary to chest pain and ST elevations)
   - Normal cardiac catheterization
   - No significant past medical history
   - Mildly dilated aortic root and ascending aorta on TTE
   - GERD, insomnia, and OSA—no active treatment or significant impact on hospitalization

3. **Complications and Comorbidities (CC and MCC):**
   - Myopericarditis increases the complexity of care and qualifies as a **CC**.
   - There is no indication that it qualifies as a major complication or comorbidity (**MCC**).

**Reasoning for the DRG Assignment:**
- Myopericarditis is the principal diagnosis and is considered a **CC**, not an MCC.
- Mildly dilated aortic root and other secondary findings do not affect DRG assignment.
- GERD, insomnia, and OSA are managed conditions that do not impact hospital resource use.

**DRG Code Assignment:**
The most likely MS-DRG for this patient, based on the provided information, is **"OTHER CIRCULATORY SYSTEM DIAGNOSES WITH CC."** This assignment reflects the presence of a complication (myopericarditis) that increases care complexity, without rising to the level of an MCC.
</think>

<answer>
**OTHER CIRCULATORY SYSTEM DIAGNOSES WITH CC**
</answer>

**Example from Differential Thinking**

**<think>**

**Analysis of the Case Details**
**Principal Diagnosis:**

- **ST-Elevations and Myopericarditis**

**Secondary Diagnoses:**

- No significant past medical history
- GERD (Gastroesophageal Reflux Disease)
- Insomnia
- OSA (Obstructive Sleep Apnea)
- HIV and Syphilis (screening recommended)

**Procedures Performed:**

- Cardiac Catheterization

**Other Relevant Factors:**

- Normal cardiac catheterization result
- Mildly dilated aortic root and ascending aorta
- Started on ibuprofen and colchicine

**Possible DRG Codes**

1. **Heart failure and shock without CC/MCC**
2. **Other circulatory system diagnoses with CC**
3. **Other circulatory system diagnoses with MCC**

**Analysis for Each DRG Code**
**1. Heart failure and shock without CC/MCC**
*Why this DRG code might be appropriate:*

- The patient was admitted with ST-elevations, which can be indicative of heart issues.
- Mild dilation of the aortic root and ascending aorta could suggest early signs of heart failure.
- Started on medications for pain and inflammation, which are common in heart failure management.

*Why it might not be appropriate:*

- Cardiac catheterization was normal, which doesn't support a diagnosis of heart failure.
- No history of heart failure or shock—only myopericarditis and mild aortic dilation.
- GERD, insomnia, and OSA are unrelated to heart failure.

**2. Other circulatory system diagnoses with CC**
*Why this DRG code might be appropriate:*

- ST-elevations and myopericarditis are circulatory system issues.
- GERD (a CC) increases complexity of care and resource utilization.

*Why it might not be appropriate:*

- Condition managed with ibuprofen and colchicine—does not indicate severe complications.
- Absence of heart failure or severe circulatory system condition to warrant a more specific DRG.

**3. Other circulatory system diagnoses with MCC**
*Why this DRG code might be appropriate:*

- ST-elevations and myopericarditis are significant circulatory issues.
- GERD (a CC) could be misclassified as MCC if chronic impact is overestimated.

*Why it might not be appropriate:*

- Condition managed conservatively—not indicative of MCC-level complexity.
- No evidence of heart failure or other major circulatory complications.

**Conclusion**
**Most Likely DRG Code: Other circulatory system diagnoses with CC**
**Reasoning:**

- Myopericarditis with ST-elevations is a significant circulatory condition.
- GERD, as a CC, increases care complexity but not to the level of MCC.
- No evidence of heart failure or other severe conditions to justify a different DRG code.

Based on the provided information, the most appropriate DRG code is **Other circulatory system diagnoses with CC**, as it best reflects the patient's primary condition and the presence of a CC (GERD) that increases complexity of care.
**</think>**

**<answer>**
**OTHER CIRCULATORY SYSTEM DIAGNOSES WITH CC**
**</answer>**

