# OpenReview forum: "Reinforcement Learning for Out-of-Distribution Reasoning in LLMs: An Empirical Study on Diagnosis-Related Group Coding"
_NeurIPS.cc/2025/Conference — NeurIPS 2025 spotlight_

### Official Review · Reviewer_RJgS · 2025-06-29

**Clarity:** 3
**Significance:** 4
**Originality:** 4
**Rating:** 4
**Confidence:** 4

**Summary:**

This paper introduces DRG-SAPPHIRE, a model for automated DRG coding from clinical notes using large-scale reinforcement learning. Built on Qwen2.5-7B and trained with GRPO, the model addresses the out-of-distribution nature of DRG coding, where LLMs have limited exposure to private clinical data. The authors systematically investigate the relationship between SFT and RL performance, finding that RL effectiveness scales logarithmically with SFT examples. DRG-SAPPHIRE achieves SOTA on MIMIC-IV and generates physician-validated reasoning for code assignments. The study reveals domain-specific challenges in applying RL to medical tasks, including preference for Answer-First cognitive patterns and sensitivity to KL divergence.

**Questions:**

Please see the questions mentioned in the weaknesses.

**Ethical Concerns:**

["NO or VERY MINOR ethics concerns only"]

**Final Justification:**

The authors have successfully addressed all three concerns raised in my initial review. First, they provided a compelling justification for prioritizing a large-scale empirical study over deeper mechanistic investigations of individual phenomena. Their explanation clarified that the observed findings about completion length contraction and answer-first patterns align with recent work in other domains. Second, the formatting inconsistency has been corrected. Third, while the study focuses on a single task, the authors effectively contextualized their findings within broader research that supports their conclusions about SFT-RL trade-offs.

Since there are no major problems that need to be addressed, I continue to suggest weak acceptance. Because of the limited task and scope that this paper's methodology may be applicable to, as well as the absence of thorough mechanistic investigations, I do not raise my score.

**Limitations:**

yes

**Paper Formatting Concerns:**

The paper contains formatting inconsistencies, such as "Kl Decay" (lowercase 'l') in Table 1, which should be "KL Decay".

**Quality:**

4

**Strengths And Weaknesses:**

# Strenghs
1. The proposed method achieves new state-of-the-art performance on the MIMIC-IV benchmark, significantly outperforming proprietary reasoning models including GPT-4o.
2. The finding that RL performance scales logarithmically with SFT sample size, and that stronger SFT foundations lead to better post-GRPO results, provides valuable guidance for resource allocation in training reasoning models for OOD tasks.
3. The ablation study systematically examines multiple GRPO variants, reward shaping strategies, cognitive behavior interventions, and adaptive learning approaches, providing thorough insights into RL training for medical coding tasks.
4. The paper identifies unique challenges in medical domains that contrast with mathematical reasoning tasks, such as completion length contraction during training and the superiority of Answer-First over CoT-First patterns, contributing to research in this community.
# Weaknesses

1. Several interesting findings lack deeper investigation or further experiments to validate them. For instance, the reason of completion length contraction phenomenon, the superiority of Answer-First reasoning despite conventional wisdom favoring CoT-First approaches, and the hypothesis that denser rewards lead to premature convergence to local optima are presented without sufficient mechanistic explanation or validation.

2. The paper contains formatting inconsistencies, such as "Kl Decay" (lowercase 'l') in Table 1.

3. The central claim that RL effectiveness is "fundamentally constrained by the domain knowledge encoded in the base model" and that "scaling SFT may be more effective and computationally efficient than scaling RL alone" is based solely on DRG coding experiments. This conclusion requires validation across multiple medical tasks and datasets to establish its broader applicability to OOD reasoning tasks.

---

> ### Author Rebuttal · Authors · 2025-07-30
>
> We thank the reviewers for their thoughtful and positive feedback. We are encouraged that they found our motivation and idea clear (R2), novel (R1, R2), and important (R3, R4). We are particularly grateful that our experiments were unanimously found to be comprehensive and thorough (R1, R2, R3, R4). The paper was also praised for being well-written and structured (R2, R3), and the code release was noted as a major plus (R3). We are pleased that reviewers recognized that our work provides valuable guidance for resource allocation in training reasoning models for out-of-distribution (OOD) tasks (R1, R3, R4). Below, we address the specific concerns and suggestions raised by the reviewer.
>
> ## 1. Mechanistic explanation or validation on various findings from our study
> We thank the reviewer for highlighting these insightful observations and agree that a deeper mechanistic investigation into each would be highly valuable. In this work, we aimed to strike a balance between the depth and breadth of our exploration. Our primary objective was to conduct a large-scale empirical study on the optimal resource allocation between SFT and RL for a challenging OOD task, a focus which required comprehensive ablations and experiments that consumed the majority of our computational resources. This necessarily limited the scope for dedicated studies into all of the secondary phenomena we observed. Consequently, we present findings—such as the completion length contraction, the surprising efficacy of the 'Answer-First' pattern, and the hypothesis that denser rewards lead to local optima—as important, empirically-grounded early observations. While a full mechanistic validation of each was beyond the scope of this paper and would deserve a dedicated study, we believe they are valuable contributions in their own right, revealing unique dynamics of RL in this domain and providing a strong foundation for future exploration.
>
>
> Nevertheless, we provide a brief discussion on some of the findings as below, in combination with some very recent work from general domains that share similar observations.
> * **Length Contraction**: For DRG assignment, the reasoning process is structurally straightforward: identify the principal diagnosis → extract relevant secondary diagnoses → assess CC/MCC status → assign the DRG (Sec. 3.1). While domain knowledge is nontrivial, the sequence of steps is generally fixed, unlike mathematical problems where solution strategies vary considerably. Hence, it is intuitive that a well-trained LLM may converge toward shorter, more structured outputs. Indeed, Cheng et al. (2025) found that response length increases correlate with performance gains only in certain domains; notably, length contraction is observed in Code, Logic, and Tabular tasks [1]. Moreover, another study shows that RL-trained reasoning models can learn to bypass verbose CoT by internalizing the reasoning process while still producing correct answers [2].
> * **Preference for Answer-First**: Our observation that answer-first prompting yields superior performance likely reflects how LLMs encode and retrieve latent knowledge. These findings align with recent studies, which suggest that CoT and extended reasoning may not always be necessary for reasoning models, and a “no-thinking” pattern can sometimes yield better performance [3, 4]. In addition, presenting the answer early may help the model condition its explanation generation on a confident anchor, avoiding hallucinated or self-justifying CoT paths. This hypothesis aligns with the work showing that answer-first prompting can expose inconsistencies in reasoning and even outperform conventional CoT-first strategies in certain settings [5].
>
> Reference:
> 1. Revisiting Reinforcement Learning for LLM Reasoning from A Cross-Domain Perspective.
> 2. Think or Not? Selective Reasoning via Reinforcement Learning for Vision-Language Models.
> 3. Reasoning models can be effective without thinking.
> 4. Reasoning models don’t always say what they think.
> 5. Order Matters in Hallucination: Reasoning Order as Benchmark and Reflexive Prompting for Large-Language-Models.
>
> ## 2. Formatting inconsistency
> Thank you for your sharp eye and for pointing this out. We will fix the "Kl Decay" typo and ensure all formatting is consistent throughout the final camera-ready version.
>
> ## 3. Necessity of cross-domain validation
> We thank the reviewer for the valuable feedback on generalizability, and we agree that broader validation is an important next step. Our study was intentionally designed as a deep, empirical investigation into a single, highly complex OOD task—DRG coding—to enable a rigorous analysis that would be difficult to replicate with the same depth across multiple domains. This focused approach allowed for the extensive experiments on SFT-RL data trade-offs (Sec 5.2), detailed ablations (Sec 5.3), and investigation of RL prerequisites (Sec 5.4) that led to our central finding: a clear, log-linear scaling relationship between the volume of SFT data and subsequent RL performance. While based on one task, this result provides strong, empirical evidence for our hypothesis that for complex OOD tasks, RL effectiveness is fundamentally constrained by the domain knowledge infused via SFT. We therefore position our claim as an important finding that provides strong motivation for future work to test this hypothesis across other domains, as noted in our limitations (Sec E).
>
> Below, we provide a brief discussion on some very recent work from other tasks and domains that resonates with our conclusion:
> * **No New "Knowledge" from RL?**: An ongoing and pertinent debate is whether RL truly incentivizes new reasoning capabilities beyond those already encoded in the base model. Yue et al. (2025) conducted a rigorous analysis across **mathematical, coding, and visual reasoning benchmarks**, concluding that RL does not induce novel reasoning patterns beyond those already present in the pretrained model [1].
> * **Hybrid SFT and RL Training is Needed**: Lu et al. (2025) analyzed training dynamics on **complex reasoning tasks**, observing that RL excels at maintaining and improving performance within a model’s existing capabilities, whereas SFT is more effective for making progress on questions beyond the model’s current scope [2].
> * **Medical Domain Evidence**: Zuo et al. (2025), analyzing Qwen-7B models across **medical question-answering** and mathematical tasks, noted that medical tasks demand much richer factual/domain knowledge compared to mathematical tasks, which emphasize logical reasoning [3]. They concluded that in medical domains, SFT remains important due to the necessity of detailed domain knowledge, while RL enhances reasoning accuracy by pruning inaccurate or irrelevant reasoning paths.
>
> In summary, our core conclusion is reinforced by these very recent cross-domain studies. **While these papers provide corroborating evidence, our work is distinguished by its systematic and comprehensive investigation of the SFT-RL trade-off, offering a granular, quantitative analysis of this critical relationship.**
>
> Reference:
> 1. Does Reinforcement Learning Really Incentivize Reasoning Capacity in LLMs Beyond the Base Model?
> 2. Learning What Reinforcement Learning Can’t: Interleaved Online Fine-Tuning for Hardest Questions.
> 3. Knowledge or Reasoning? A Close Look at How LLMs Think Across Domains.

---

> > ### Comment · Reviewer_RJgS · 2025-08-05
> >
> > Thank you for your response, which addressed most of my concerns. Based on your explanation, I agree it is appropriate to prioritize the large-scale empirical study of this OOD task and leave deeper mechanistic investigations for future work (Q1 resolved). The formatting issue has been corrected (Q2 resolved), and the key findings have been independently validated by concurrent work (Q3 resolved).
> >
> >
> > By the way, do you have any plans for next steps?

---

> > > ### Author Response · Authors · 2025-08-05
> > >
> > > We thank the reviewer for the kind reply and are pleased that our rebuttal has addressed most of the concerns.
> > >
> > > Regarding future work, we have submitted an Institutional Review Board (IRB) application to validate and implement our model in real-world clinical settings. Through this, we aim to evaluate two key aspects:
> > >
> > > **Model Performance Validation**: Following a reviewer's suggestion, we performed a preliminary evaluation on an internal dataset of 2,500 real-world cases. DRG-Sapphire achieved an accuracy of 53.6% on this dataset, compared to 54.8% on MIMIC, suggesting robustness beyond the MIMIC corpus. While we acknowledge that this dataset is relatively small, the full IRB protocol will enable a large-scale validation of the model's performance in a production environment.
> > >
> > > **Healthcare Delivery Impact**: We plan to assess how our model integrates into clinical workflows, recognizing that acceptable fault tolerance varies by application. For instance, high-stakes tasks like automated billing require extremely low error rates, whereas operational or educational tools (e.g., Appendix D.1) may permit more flexibility. This study will allow us to formally evaluate these factors and quantify the model's real-world impact.
> > >
> > > Beyond above, we are also interested in advancing RL with verifiable rewards through the following directions:
> > >
> > > 1. Approaches to effectively manage entropy and prevent entropy collapse [1,2,3].
> > >
> > > 2. Cross-domain and prolonged RL training to activate capabilities not present in the base model [4,5].
> > >
> > > 3. Strategies for effective curriculum learning beyond difficulty-based filtering [6].
> > >
> > > References:
> > > 1. The Entropy Mechanism of Reinforcement Learning for Reasoning Language Models.
> > > 2. Beyond the 80/20 Rule: High-Entropy Minority Tokens Drive Effective Reinforcement Learning for LLM Reasoning.
> > > 3. Do Not Let Low-Probability Tokens Over-Dominate in RL for LLMs.
> > > 4. Revisiting Reinforcement Learning for LLM Reasoning from A Cross-Domain Perspective.
> > > 5. ProRL: Prolonged Reinforcement Learning Expands Reasoning Boundaries in Large Language Models
> > > 6. QuestA: Expanding Reasoning Capacity in LLMs via Question Augmentation.

---

> > > > ### Comment · Reviewer_RJgS · 2025-08-09
> > > >
> > > > I appreciate the authors’ detailed and thoughtful reply to my questions. They have addressed my concerns. I recommend adding the new results to the paper. My score remains at weak acceptance.

---

### Official Review · Reviewer_4M3r · 2025-07-01

**Clarity:** 4
**Significance:** 3
**Originality:** 3
**Rating:** 5
**Confidence:** 4

**Summary:**

This paper explores the application of large-scale reinforcement learning (RL) for out-of-distribution (OOD) reasoning in large language models (LLMs), focusing on automated Diagnosis-Related Group (DRG) coding—a critical task for hospital operations and reimbursement. The authors introduce DRG-SAPPHIRE, built on Qwen2.5-7B, which is fine-tuned using supervised fine-tuning (SFT) followed by RL with Group Relative Policy Optimization (GRPO) and rule-based rewards. The study provides an extensive empirical evaluation of GRPO variants, adaptive learning strategies (e.g., curriculum and staged learning), and reward shaping techniques. Results demonstrate state-of-the-art (SOTA) accuracy on the MIMIC-IV dataset and show that DRG-SAPPHIRE generates physician-validated reasoning, improving transparency and explainability. A key finding is that RL performance scales approximately linearly with the logarithm of SFT sample size, reinforcing that strong base model knowledge is essential for success in OOD domains.

**Questions:**

See my comments above.

**Ethical Concerns:**

["NO or VERY MINOR ethics concerns only"]

**Final Justification:**

Again, I think this is a solid paper. I appreciate that the authors made significant efforts to support their claims. I recommend accepting this manuscript.

**Limitations:**

The manuscript would benefit from a dedicated limitations discussion (e.g., on generalizability beyond MIMIC-IV, reliance on high-quality SFT data, scalability of training cost).

**Paper Formatting Concerns:**

No format concerns.

**Quality:**

4

**Strengths And Weaknesses:**

> Strengths

- Well written and structured: The paper is clear, professional, and accessible despite the technical complexity of the topic.
- Important and timely problem: The focus on DRG coding — a high-impact, real-world OOD task in healthcare — is highly relevant and valuable.
- Comprehensive evaluation: The authors perform rigorous analyses of GRPO, including ablation studies, enhancements (e.g., dynamic resampling, KL decay), and data allocation strategies between SFT and RL.
- Informative insights: The investigation into RL’s limitations and failure modes provides a useful reference for the community.
- Did a great job on sharing their codes.

> Weakness
- While the empirical exploration is thorough, the main insight — that strong RL performance depends on prior knowledge infusion — is expected and aligns with existing understanding (i.e., the “bitter lesson” that scaling data or pretraining matters more than algorithmic tweaks).
- Lack of coverage of related recent work: I personally love the idea of rule-based reward as it will be a huge cost saving but I also like to see you have more discussion on other people's work on rule-based rewards. For instance:
   Mu, Tong, et al. "Rule based rewards for language model safety." arXiv preprint arXiv:2411.01111 (2024).
   Sun, Shenghuan, et al. "Dr-LLaVA: Visual Instruction Tuning with Symbolic Clinical Grounding." arXiv preprint arXiv:2405.19567 (2024).
- Missing limitations section: The manuscript would benefit from a dedicated limitations discussion (e.g., on generalizability beyond MIMIC-IV, reliance on high-quality SFT data, scalability of training cost).

---

> ### Author Rebuttal · Authors · 2025-07-30
>
> We thank the reviewers for their thoughtful and positive feedback. We are encouraged that they found our motivation and idea clear (R2), novel (R1, R2), and important (R3, R4). We are particularly grateful that our experiments were unanimously found to be comprehensive and thorough (R1, R2, R3, R4). The paper was also praised for being well-written and structured (R2, R3), and the code release was noted as a major plus (R3). We are pleased that reviewers recognized that our work provides valuable guidance for resource allocation in training reasoning models for out-of-distribution tasks (R1, R3, R4). Below, we address the specific concerns and suggestions raised by the reviewer.
> ## 1. Question about the insight relates to the “Bitter lesson”
> * Thank you for your insightful feedback. While our conclusion aligns with Sutton’s "bitter lesson," we hope our work offers a timely and empirically grounded clarification for the domain of complex, out-of-distribution (OOD) tasks.
> * Following successes of reasoning models like DeepSeek-R1, a prevailing narrative has been to "scale RL," **leaving a critical question unanswered: what, precisely, should be scaled?** Our work addresses this for complex, OOD tasks where the answer is not straightforward. We demonstrate that naively scaling RL (in terms of data and compute) on a base model fails entirely for DRG coding, as vanilla models were unable to produce correct outputs with RL alone.
> * Through rigorous experimentation, we conclude that for OOD challenges, knowledge infusion is the critical component to scale. Our findings suggest that scaling SFT is often more effective and computationally efficient than scaling RL alone. In this light, our work encourages the community to more carefully consider what is being scaled when applying RL. We hope this clarification serves as a meaningful and timely contribution, offering an evidence-based explanation and a clear path forward for the many practitioners working to apply RL to domain-specific OOD tasks.
>
> ## 2. Additional literatures
> Thank you for suggesting these insightful references. We would be happy to include both papers as related work in the camera-ready version. The first paper from OpenAI presents an effective approach that uses fine-grained, composable rules and an LLM grader to create a hybrid reward signal, balancing model safety and usefulness [1]. We will also highlight the novel framework in the second paper, which employs symbolic representations of clinical reasoning to ground vision-language models (VLMs) in medical knowledge and uses a symbolic reward function to evaluate VLM responses for individual correctness and clinical validity [2].
>
> Reference:
> 1. Rule based rewards for language model safety.
> 2. Dr-LLaVA: Visual Instruction Tuning with Symbolic Clinical Grounding.
>
> ## 3. Missing limitations section
> * We thank the reviewer for the helpful suggestions and wish to clarify that a limitations section is provided in Appendix Sec E. In the camera-ready version, we will expand this section to incorporate your valuable points on scalability and SFT data.
> * Of note, following another reviewer’s kind suggestion and with IRB approval, we conducted an additional evaluation using an internal dataset of 2,500 real-world cases from a healthcare institution. DRG-Sapphire achieved an accuracy of 53.6% on this internal dataset, compared to 54.8% on MIMIC, demonstrating its robustness beyond the MIMIC corpus. We also submitted an IRB application for validating and implementing our model in real-world clinical settings at large scale.

---

> > ### Comment · Reviewer_4M3r · 2025-08-06
> >
> > I thank the authors for their detailed responses. I appreciate how you addressed my concerns and would encourage reflecting these points in the discussion section of the paper. My score remains unchanged.

---

> ### Author Response · Authors · 2025-08-07
>
> We appreciate the reviewer's positive feedback. We are pleased that we could address your concerns and would be happy to answer any further questions.

---

### Official Review · Reviewer_A6Eg · 2025-07-02

**Clarity:** 3
**Significance:** 3
**Originality:** 3
**Rating:** 5
**Confidence:** 3

**Summary:**

The authors propose a GRPO based algorithm for training LLMs to predict and reason in domains with out of distribution data compared to their training data. Specifically, they focus on Diagnosis-Related Group (DRG) codes prediction from clinical notes which involves identification of the principal diagnosis, principal procedure, complications or comorbidities, and major complications or comorbidities. They present their trained model called DRG-Sapphire, which is a fine-tuned Qwen2.5-7B-Instruct model. This model achieves SoTA results in DRG code prediction as well as provides a human-interpretable explanation for the DRG code assignment. The authors perform extensive ablations and provide detailed analysis of various algorithmic variants which they considered in developing DRG-Sapphire. The authors also provide insights such as the importance of SFT prior to RL training for new data domains and also a scaling law for RL performance with respect to SF data size.

**Questions:**

1. Some analysis of evidence on LLMs struggling with DRG coding would be useful. Specifically, what are the kinds of examples where other LLMs fail where DRG-Sapphire does well and vice-versa.
2. According to the authors, what is the reason for completion length contraction?
3. The authors mention a “Bitter-lesson” in their conclusion. Is this realization from a single domain example? Can this be attributed just to novelty of data or also to the complexity of the task?
4. What was the accuracy of the final SFT dataset used? Does SFT dataset produced by a larger (better) model such as Qwn2.5-14B-Instruct improve all downstream RL performance?
5. Can this process be done iteratively? Generate new SFT data using DRG-Sapphire and continue the process?
6. Was LoRA tested? Are all observations consistent even when LoRA is used?
7. Why did the authors choose a pure-online variant of GRPO?

**Ethical Concerns:**

["NO or VERY MINOR ethics concerns only"]

**Final Justification:**

Based on my interaction with the authors and the comments from the other reviewers, I would like to retain my score. All my major concerns have been addressed.

**Limitations:**

Have been discussed except for impact due to misclassification by DRG-Sapphire. Some discussion regarding that would be useful.

**Paper Formatting Concerns:**

None.

**Quality:**

3

**Strengths And Weaknesses:**

Strengths:
1. The paper is very well written and the concepts, problem and the solution approach are clearly explained.
2. The authors provide algorithmic improvements such as a novel dynamic resampling strategy
3. The authors conduct a thorough exploration and experimentation with various algorithmic approaches presented in literature and present  extensive ablation studies.

Weaknesses:
1. I did not see experiments using in-context learning examples with large models, which have not been specifically trained with DRG data.
2. I also could not find any discussion on impact of erroneous DRG code assignment.
3. Please also see my questions for additional points.

---

> ### Author Rebuttal · Authors · 2025-07-30
>
> We thank the reviewers for their thoughtful and positive feedback. We are encouraged that they found our motivation and idea clear (R2), novel (R1, R2), and important (R3, R4). We are particularly grateful that our experiments were unanimously found to be comprehensive and thorough (R1, R2, R3, R4). The paper was also praised for being well-written and structured (R2, R3), and the code release was noted as a major plus (R3). We are pleased that reviewers recognized that our work provides valuable guidance for resource allocation in training reasoning models for out-of-distribution (OOD) tasks (R1, R3, R4). Below, we address the specific concerns and suggestions raised by the reviewer.
>
> ## 1. Question about in-context learning examples
> * Due to the large DRG search space (over 700 distinct codes) and the base model's lack of DRG-specific knowledge (Fig. 1A), we found in-context learning (ICL) examples insufficient for effective knowledge injection. It is not feasible to cover the full code space and their associated clinical notes within a few ICL examples. While increasing the number of ICL examples could improve coverage, it would also lead to a quadratic increase in inference cost.
> * A RAG-based approach for selecting examples would also be problematic for two reasons: (1) Clinical notes with similar semantic features can map to very different DRG codes, making retrieval unreliable. (2) The retrieved examples would constrain the model's output to a small subset of DRG codes, limiting its ability to reason over the entire code space. For these reasons, we chose to apply SFT to infuse domain-specific knowledge before RL.
>
> ## 2. Suggestion to add error analysis
> We conducted additional error analyses as kindly suggested by the reviewer.
> * We identified 9,537 cases consistently misclassified by all three checkpoints from different training stages: the cold-start SFT (SFT model in Fig. 5A), the small-scale GRPO (GRPO model in Fig. 5A), and the final GRPO-Sapphire model. As shown in the table below, even in these challenging cases, the model improved progressively during training, becoming notably "less wrong."
>
> | Model                  | % Wrong Principal Diagnosis | % Wrong CC/MCC | % Both Wrong |
> |-|-|-|-|
> | Coldstart SFT          | 85.1%                       | 64.2%                  | 47.4%         |
> | Coldstart SFT + GRPO   | 80.4%                       | 63.1%                  | 41.1%         |
> | DRG-Sapphire           | 74.9%                       | 62.7%                  | 33.6%         |
>
> * Additionally, a physician expert conducted a detailed manual error analysis on 100 randomly selected cases misclassified by the final GRPO-Sapphire model, applying the same error taxonomy used in DRG-LLaMA.
>
> | Error Category                                        | %     |
> |------------------------------------------------------|-------|
> | Difficulty selecting correct base DRG                | 32.0% |
> | Erroneous CC/MCC                                     | 25.0% |
> | Potential incorrect DRG label                        | 13.0% |
> | Inadequate clinical concept extraction               | 12.0% |
> | Info needed for DRG prediction unavailable           | 10.0% |
> | Other (e.g., procedure instead of medical DRG)       |  8.0% |
>
> * The physician expert also marked 15% of these cases as ambiguous, noting that the correct DRG was unclear based on DRG guidelines, for example due to multiple potential principal diagnoses. Collectively, these results underscore the inherent complexity of DRG assignment. Interestingly, the identification of potentially incorrect DRG labels highlights a promising potential use case for our model in detecting medical coding errors.
>
> * When reviewing cases where other proprietary LLMs fail, we observed that most failures stem from a lack of DRG-specific knowledge, highlighting the OOD nature of the task. For example, these models may output invalid DRG codes or struggle to determine whether a secondary condition qualifies as a CC or MCC.
>
> ## 3. Question about impact of erroneous DRG code assignment
> We thank the reviewer for raising this important point. After consulting with physician leaders, we agree that the impact of errors is highly use case-dependent. For high-stakes applications like automated billing, the error tolerance must be extremely low due to potential financial consequences. In contrast, for operational or educational tools—such as the one in Appendix D.1 that helps physicians understand DRG assignments—a higher degree of fault tolerance may be acceptable. To rigorously assess these factors in practice, we have submitted an IRB application to prospectively validate the model's performance and impact within a healthcare system. This will enable a formal evaluation from a healthcare delivery perspective.
>
> ## 4. Question about the reason of completion length contraction
> We hypothesize that the completion length contracts because DRG assignment is a highly structured reasoning task, allowing the model to learn a more concise and efficient solution path. Due to word limits, we refer the reviewer to our response to RJgS (Q1) for a full analysis with supporting references.
>
> ## 5. Question about cross-domain generalization of the “Bitter-lesson”
> Our conclusion is grounded in comprehensive experiments on the OOD task of DRG assignment, and we suggest these findings are likely to generalize to other OOD tasks where the model has minimal pre-training exposure. We provide a detailed discussion with supporting literature in our response to Reviewer RJgS (Q3).
>
> ## 6. Performance when training on SFT dataset produced by 14B model
> This is a great thought, and we have conducted additional experiments as the reviewer suggested.
>
> | Setting             | Experiment | 7B    | 14B   |
> |-|-|-|-|
> | 5% SFT, 95% RL      | SFT        | 25.0% | 21.8% |
> |                     | +GRPO      | 38.7% | 34.6% |
> | 75% SFT, 25% RL     | SFT        | 38.5% | 35.7% |
> |                     | +GRPO      | 47.4% | 43.8% |
>
> Interestingly, RL training based on SFT data produced by the 14B model did not improve performance. We suspect this may be due to a teacher–student capacity gap: the smaller 7B student model struggled to imitate the complex style and output distribution of the larger model, resulting in suboptimal policy initialization that hindered RL optimization.
>
> ## 7. Question about SFT dataset
> We are happy to clarify this important detail. The SFT dataset contains two parts: (1) the true DRG label from MIMIC, which is always correct, and (2) the CoT reasoning trace. For the CoT trace, we differentiate between its **factuality** (e.g., whether a condition is CC vs. MCC) and its **cognitive pattern** (the reasoning logic). We do not expect the initial CoT traces to be factually perfect, as the Qwen base model lacks DRG-specific knowledge. Instead, it is crucial that the cognitive pattern is correct. Through prompt engineering (Sec. H), we introduce diverse and valid cognitive patterns for the model to learn during SFT (lines 162–165). This provides a sound structure for the RL phase to build upon and learn factuality.
>
> ## 8. Question about iterative training
> We agree that training can be done iteratively, either via repeated SFT/RL cycles as the reviewer suggests or through multiple policy updates per generation. However, there are two key considerations for such a process:(1) RL training risks entropy collapse as training progresses [1], necessitating careful entropy management to ensure training success [2, 3]. (2) If the SFT dataset from the latest checkpoint does not introduce new knowledge, iterative learning may be ineffective. A recent study proposes a compelling approach, introducing partial solutions during iterative training to simplify problem difficulty and reshape the learnability landscape [4]. This increases the likelihood of discovering correct trajectories, leading to improved RL training outcomes.
>
> Reference:
> 1. The Entropy Mechanism of Reinforcement Learning for Reasoning Language Models.
> 2. Beyond the 80/20 Rule: High-Entropy Minority Tokens Drive Effective Reinforcement Learning for LLM Reasoning.
> 2. Do Not Let Low-Probability Tokens Over-Dominate in RL for LLMs.
> 3. QuestA: Expanding Reasoning Capacity in LLMs via Question Augmentation.
>
> ## 9. Question about LoRA
> We did not test LoRA because the primary bottleneck in our RL work, as in many RL with verifiable rewards studies, was the computational cost (inference time) of generating rollouts, not GPU memory. Since memory was not the constraint, we opted for full-parameter training to obtain a cleaner evaluation of our method, thereby avoiding the complexity and potential confounding variables of LoRA-specific hyperparameters (e.g., rank selection).
>
> ## 10. Question about pure-online GRPO
> We chose the pure-online variant of GRPO for several reasons.
> * Our approach aligns with the original GRPO implementation from DeepSeek-Math [1] which is pure-online.
> * This decision is supported by growing evidence that on-policy RL can yield better performance than off-policy settings [2, 3]. Specifically, off-policy learning has been shown to accelerate entropy collapse [2]. Furthermore, recent work by Hao et al. (2025) identifies potential overfitting issues associated with off-policy learning [3].
> * From an experimental design perspective, this choice reduces the number of hyperparameters to tune, simplifying the setup and leading to a cleaner analysis.
>
> Reference:
> 1. DeepSeekMath: Pushing the Limits of Mathematical Reasoning in Open Language Models.
> 2. Skywork Open Reasoner 1 Technical Report.
> 3. On-Policy RL with Optimal Reward Baseline.

---

> ### Comment · Reviewer_A6Eg · 2025-08-06
>
> I thank the authors for their detailed responses and the additional experiments. Most of my questions/concerns have been addressed. I am not sure about the conclusions derived by the authors with respect to the observations pertaining to experiments in "Performance when training on SFT dataset produced by 14B model". However, I do not have any alternate hypothesis as well in this regard.

---

> > ### Author Response · Authors · 2025-08-07
> >
> > We thank the reviewer for the kind response and are glad to hear that our rebuttal addressed your concerns. Please let us know if you have any further questions, and we would be happy to answer them.

---

### Official Review · Reviewer_DCiq · 2025-07-19

**Clarity:** 3
**Significance:** 2
**Originality:** 2
**Rating:** 4
**Confidence:** 4

**Summary:**

This paper explores the use of reinforcement learning methods represented by GRPO for the DRG coding task and introduces the DRG-Sapphire model, which achieves SOTA performance.
The authros design and conduct extensive ablation experiments, providing a comprehensive analysis of the effectiveness and interactions of RL and SFT in this task.

**Questions:**

see above

**Ethical Concerns:**

["NO or VERY MINOR ethics concerns only"]

**Final Justification:**

I acknowledge the authors' responses, and hence raise my score towards acceptance.

**Limitations:**

yes

**Quality:**

3

**Strengths And Weaknesses:**

Strengths

1. The problem addressed is highly useful in real world. The authors are (probably) the first to apply advanced reasoning models and GRPO methods to this task.
2. The ablation studies are astonishingly rich, providing valuable insights for the community to understand the dynamics of RL trained LLMs.
3. There are several novel and interesting settings, such as the exploration of cognitive behaviors and the investigation of how to allocate data between SFT and RL under a fixed budget.


Weaknesses

Major

1. My major concern lies in the exact source of the performance improvement: the baseline models are highly likely to lack exact knowledge of the coding system and thus fail to generate correct codes. The performance improvement may then stem from the adaption to the coding system and the text distribution, rather than enhanced capacities.
2. Only one train and test set is used, which come from the same distribution, which further indicates that the performance improvement comes from domain adaption.
3. There lacks error analysis for baseline models or the abalted models, which may reveal deeper understanding of the model dynamics.
4. Also, it is not clearly explained how rewards are determined. If rewards are based on naive string match, then the model may be simply memorizing the vocabulary.   Combining the above weaknesses, I’m quite confident that your model is simply memorizing the coding vocabulary and adapting to the MIMIC corpus distribution, which undermines the claim of enhanced medical reasoning capacities.

Minor

1. Dynamic resampling has been proposed in numerous existing studies.
2. The construction of SFT data using Qwen2.5-7B is insufficient. The human evaluation is vague and lacks quantitative metrics, failing to validate the data quality.
3. The improvement over the previous SOTA (DRG-Llama) is marginal.

---

> ### Author Rebuttal · Authors · 2025-07-30
>
> We thank the reviewers for their thoughtful and positive feedback. We are encouraged that they found our motivation and idea clear (R2), novel (R1, R2), and important (R3, R4). We are particularly grateful that our experiments were unanimously found to be comprehensive and thorough (R1, R2, R3, R4). The paper was also praised for being well-written and structured (R2, R3), and the code release was noted as a major plus (R3). We are pleased that reviewers recognized that our work provides valuable guidance for resource allocation in training reasoning models for out-of-distribution (OOD) tasks (R1, R3, R4). Below, we address the specific concerns and suggestions raised by the reviewer.
>
> ## 1. Question on the exact source of the performance improvement
> We address this important question as follows:
> * **TL;DR**: Initial SFT (or “adaptation,” as the reviewer puts it) is necessary but NOT sufficient for improving LLM performance on OOD tasks. Reinforcement learning with verifiable rewards (RLVR) further enhances reasoning and generalization beyond what SFT alone provides.
> * We agree with the reviewer that the baseline models indeed lack relevant knowledge about medical coding. This is precisely the main focus of our paper: investigating how RLVR can be effectively applied to OOD tasks. We demonstrate that RLVR alone is insufficient for injecting domain-specific knowledge (Fig 10A); thus, initial SFT ("domain adaptation," as noted by the reviewer) is essential. However, after this warming-up phase, RLVR further enhances performance, indicating that RLVR offers benefits beyond simple domain adaptation.
> * We also agree with the reviewer that the "domain adaptation," such as memorizing DRG vocabulary and adapting to the MIMIC corpus, is important but represents only a partial view of knowledge infusion in our complex task. Medical coding tasks vary in complexity; for instance, ICD coding may straightforwardly match terms like "bladder infection" to "cystitis" when MEAT criteria (Monitoring, Evaluating, Assessing, and Treating) are satisfied. However, DRG coding is significantly more intricate. A clinical note mentioning "cystitis" may or may not correspond to the DRG code "KIDNEY AND URINARY TRACT INFECTIONS," depending on principal diagnosis designation and competing conditions or procedures. Moreover, the severity and complications (e.g., simple infection versus septic shock) influence whether "cystitis" is categorized as a CC or MCC. As detailed in lines 42-45, this complexity highlights the importance of reasoning capabilities in our model.
> * Throughout the RLVR process, we clearly observed advancements in the critical reasoning capabilities required for accurate DRG assignment. Specifically, the model learned to identify the correct principal diagnosis and assess whether secondary conditions meet CC/MCC criteria based on clinical complexity (see Sec I for examples). Additional error analysis presented below (Q3) further support this observation. **Crucially, these reasoning capabilities emerged during the RLVR stage, after domain adaptation via SFT was complete, thus reflecting learning beyond mere supervised memorization.**
> * When discussing memorization and overfitting to a data distribution, it is important to distinguish between RL and SFT. Recent work has demonstrated that RL in LLMs can achieve much better generalization than SFT and moves beyond memorization, with findings suggesting that “SFT Memorizes, RL Generalizes” [1]. Additionally, RL-tuned math reasoning models generalize well across diverse domains and tasks [2].
> * Additionally, below we provide robust DRG-Sapphire performance on our internal dataset, extending beyond the MIMIC corpus. Collectively, these findings strongly suggest our model has achieved deeper learning than simply memorizing coding vocabulary and adapting to the MIMIC corpus distribution.
>
> Reference:
> 1. SFT Memorizes, RL Generalizes: A Comparative Study of Foundation Model Post-training.
> 2. Does Math Reasoning Improve General LLM Capabilities? Understanding Transferability of LLM Reasoning.
>
> ## 2. Only one train and test set is used
> * We appreciate the reviewer’s valuable suggestion and, following IRB approval, conducted an additional evaluation using an internal dataset from a healthcare institution. We constructed this dataset by stratified random sampling of 2,500 real-world cases, drawing each DRG code in proportion to its frequency in the MIMIC test set to ensure comparability. DRG-Sapphire achieved an accuracy of 53.6% on this internal dataset, compared to 54.8% on MIMIC, demonstrating its robustness beyond the MIMIC corpus. We also submitted an IRB application for validating and implementing our model in real-world clinical settings at large scale.
> * To the best of our knowledge, MIMIC is the only publicly available dataset containing both clinical notes and DRG labels, and was the sole dataset used in prior work such as DRG-LLaMA. While we acknowledge the limitation of a single source, MIMIC’s training and test sets are large (230k and 28k cases, respectively) and reflect real-world patient distributions in North America, as they span over a decade of ICU and ED admissions at Beth Israel Deaconess Medical Center in Boston, Massachusetts.
>
> ## 3. A lack of error analysis
> We conducted additional qualitative and quantitative error analyses as kindly suggested by the reviewer.
> * We identified 9,537 cases consistently misclassified by all three checkpoints from different training stages: the cold-start SFT (SFT model in Fig. 5A), the small-scale GRPO (GRPO model in Fig. 5A), and the final GRPO-Sapphire model. As shown in the table below, even in these challenging cases, the model improved progressively during training, becoming notably "less wrong."
>
> | Model                  | % Wrong Principal Diagnosis | % Wrong CC/MCC | % Both Wrong |
> |-|-|-|-|
> | Coldstart SFT          | 85.1%                       | 64.2%                  | 47.4%         |
> | Coldstart SFT + GRPO   | 80.4%                       | 63.1%                  | 41.1%         |
> | DRG-Sapphire           | 74.9%                       | 62.7%                  | 33.6%         |
>
>
> * Additionally, a physician expert conducted a detailed manual error analysis on 100 randomly selected cases misclassified by the final GRPO-Sapphire model, applying the same error taxonomy used in DRG-LLaMA.
>
> | Error Category                                        | %     |
> |------------------------------------------------------|-------|
> | Difficulty selecting correct base DRG                | 32.0% |
> | Erroneous CC/MCC                                     | 25.0% |
> | Potential incorrect DRG label                        | 13.0% |
> | Inadequate clinical concept extraction               | 12.0% |
> | Info needed for DRG prediction unavailable           | 10.0% |
> | Other (e.g., procedure instead of medical DRG)       |  8.0% |
>
> * The physician expert also marked 15% of these cases as ambiguous, noting that the correct DRG was unclear based on DRG guidelines, for example due to multiple potential principal diagnoses. **Collectively, these results underscore the inherent complexity of DRG assignment.** Interestingly, the identification of potentially incorrect DRG labels highlights a promising potential use case for our model in detecting medical coding errors.
>
> ## 4. Question about reward determination
> We detailed our rule-based reward modeling in Sec A.2 (lines 460–468). For the accuracy reward, we used exact string matching. This stringent rule-based approach follows the standard practice in recent RLVR work, beginning with DeepSeek-R1. Such minimalism is key to mitigating the well-known issue of reward hacking in RL and forms a cornerstone of RLVR methodology. In early experiments, we explored more relaxed reward regimens (e.g., partial string matching). However, consistent with prior work, we observed frequent reward hacking—for example, the model would generate incorrect but verbose answers to game the reward function through partial matches. Finally, we wish to clarify that while we used string match, we assigned different scores for full versus partial DRG matches (e.g., correct principal diagnosis only or CC/MCC status only), thereby providing denser and more informative reward signals.
>
> ## 5. Novelty about dynamic resampling
> We emphasize that this work is not about introducing yet another SFT/RLVR trick. Rather, our goal is to understand how SFT and RLVR influence LLM learning in OOD tasks. To our knowledge, we are among the first to highlight several practical insights, particularly the training cost analysis in Fig. 7C, which we hope will be valuable to the community.
>
> ## 6. Question about quality of SFT dataset constructed by Qwen2.5-7B
> We are happy to clarify this important detail. The SFT dataset contains two parts: (1) the true DRG label from MIMIC, which is always correct, and (2) the CoT reasoning trace. For the CoT trace, we differentiate between its **factuality** (e.g., whether a condition is CC vs. MCC) and its **cognitive pattern** (the reasoning logic). We do not expect the initial CoT traces to be factually perfect, as the Qwen base model lacks DRG-specific knowledge. Instead, it is crucial that the cognitive pattern is correct. Through prompt engineering (Sec. H), we introduce diverse and valid cognitive patterns for the model to learn during SFT (lines 162–165). This provides a sound structure for the RLVR phase to build upon and learn factuality.
>
> ## 7. Performance improvement
> Compared to prior classification-based methods, DRG-Sapphire offers the key advantage of generating high-quality explanations for DRG assignments (Fig 4), while also achieving SOTA performance. In the medical domain, such interpretability is essential for trust and transparency. As emphasized by physician leaders (Sec D.1), these interpretable rationales are foundational for real-world deployment, enabling actionable clinical insights.

---

> ### Author Response · Authors · 2025-08-06
>
> Dear Reviewer,
>
> With the discussion period concluding soon, we hope our rebuttal has addressed your concerns, including the one about the source of performance improvement (Q1).
>
> Please let us know if you have any remaining comments; we are happy to elaborate further. We deeply appreciate the effort you have dedicated to reviewing our work.

---

### Note · Authors · 2025-08-12

Dear Reviewers and Area Chair,

We sincerely thank all reviewers for their thoughtful and constructive feedback. In these final remarks, we briefly summarize the rebuttal and discussion. For clarity, we refer to Reviewers DCiq, A6Eg, 4M3r, and RJgS as R1, R2, R3, and R4, respectively.

**Positive Feedback**

We are grateful for the encouraging assessments from all reviewers, who highlighted:
* **Motivation & Impact**: Clear motivation (R2), an important and timely problem (R1, R3), and valuable guidance for resource allocation in training reasoning models for OOD tasks (R4).
* **Experimental Rigor**: Extensive experiments (R2), “astonishingly” rich ablations (R1), and a comprehensive evaluation (R3, R4).
* **Contributions**: Novel findings (R1), informative insights (R3), and algorithmic improvements (R2).
* **Presentation Quality**: “Very well written” (R2, R3, R4).

**Clarifications in Rebuttal**
* We clarified the source of performance improvements: initial SFT is necessary but NOT sufficient for LLM enhancement, and reasoning gains occurred during the RLVR stage after domain adaptation via SFT.
* Explained why our work offers a timely clarification of the “bitter lesson” for RL: while the prevailing narrative since DeepSeek-R1 has been to “scale RL,” we show that naively scaling RL on a base model fails entirely for DRG coding and provide a practical recipe for adapting RL to OOD domains.
* We clarified experimental details, such as reward determination, and highlighted the relevant sections in the manuscript.

**Additional Experiments**
* Evaluated DRG-Sapphire on an internal dataset, confirming robustness beyond the MIMIC corpus.
* Conducted qualitative and quantitative error analyses, showing continued improvement even on the most challenging cases.
* Tested using SFT data generated by the larger 14B model.

**Addressing Reviewer Questions**
* Provided detailed, literature-backed responses, including on completion length contraction and cross-domain generalizability.
* Expanded discussion of related work.
* Outlined next steps to study the model in real-world health delivery settings.

We have carefully answered each reviewer’s comments and are pleased that R2, R3, and R4 have confirmed most of their concerns have been addressed. **While we did not get a chance to hear back from R1, we hope our detailed responses, together with the completion of all requested experiments, sufficiently address their points.**

---

### Decision · Program_Chairs · 2025-09-17

**Decision:**

Accept (spotlight)

**Comment:**

The paper had mixed results with two suggesting a borderline accept and two suggesting accept. The reviewers agree that the paper is well motivated and that the experiments are convincing. I am happy to see a paper that successfully uses RL in the medical domain and recommend acceptance.